# Charge-altering releasable transporters enhance mRNA delivery in vitro and exhibit in vivo tropism

Zhijian Li [1,9], Laura Amaya [2,3,9], Ruoxi Pi [4], Sean K. Wang [2,5], Alok Ranjan [1], Robert M. Waymouth [1], Catherine A. Blish [4,6], Howard Y. Chang [2,7] & Paul A. Wender [1,8]

The introduction of more effective and selective mRNA delivery systems is required for the advancement of many emerging biomedical technologies including the development of prophylactic and therapeutic vaccines, immunotherapies for cancer and strategies for genome editing. While polymers and oligomers have served as promising mRNA delivery systems, their efficacy in hard-to-transfect cells such as primary T lymphocytes is often limited as is their cell and organ tropism. To address these problems, considerable attention has been placed on structural screening of various lipid and cation components of mRNA delivery systems. Here, we disclose a class of charge-altering releasable transporters (CARTs) that differ from previous CARTs based on their beta-amido carbonate backbone (bAC) and side chain spacing. These bAC-CARTs exhibit enhanced mRNA transfection in primary T lymphocytes in vitro and enhanced protein expression in vivo with highly selective spleen tropism, supporting their broader therapeutic use as effective polyanionic delivery systems.

Emerging RNA technologies have enabled new and often remarkably effective strategies for therapeutic and prophylactic vaccinations[1], immunotherapies[2], and genome editing[3]. A key to the further advancement of these technologies is the introduction of more efficient and selective delivery systems for polyanions including mRNA, small interfering RNA (siRNA)[4], circular RNA (circRNA)[5], self-amplifying RNA (saRNA)[6], plasmid DNA[7], and polyanion combinations[8]. mRNA, in particular, has figured prominently in current studies as it avoids genome integration, is amenable to manufacturing and only transiently induces de novo protein synthesis in cells. Despite the enormous progress, naked mRNA is large and polyanionic, unable to efficiently cross nonpolar biological barriers such as the plasma membrane and reach the cytosol to elicit its function in vivo. To realize the full potential of mRNA therapeutics therefore requires efficient delivery technologies. In recent years, significant progress has been made in optimizing lipid nanoparticles (LNPs) for RNA delivery, as evident from the FDA approval of COVID-19 mRNA vaccines[9,10] and multiple therapeutic assets in active clinical trials[3,11,12]. However, LNPs used in clinical trials are mostly restricted to liver delivery when administered intravenously. While there are recent reports of LNP formulations for RNA delivery to the lungs[13,14], lymphatic systems[13,14], and bone marrow[15], no extrahepatic mRNA therapeutics (intravenous) have been approved thus far for clinical use, highlighting the need for novel delivery systems for organ- and cell-specific mRNA delivery.

[1]Department of Chemistry, Stanford University, Stanford, CA 94305, USA. [2]Center for Personal Dynamic Regulomes, Stanford University, Stanford, CA 94305, USA. [3]Institute for Stem Cell Biology and Regenerative Medicine, Stanford University School of Medicine, Stanford, CA 94305, USA. [4]Division of Infectious Diseases and Geographic Medicine, Department of Medicine, Stanford, CA 94305, USA. [5]Department of Ophthalmology, Stanford University School of Medicine, Stanford, CA 94305, USA. [6]Chan Zuckerberg Biohub, San Francisco, CA, USA. [7]Howard Hughes Medical Institute, Stanford University, Stanford, CA 94305, USA. [8]Department of Chemical and Systems Biology, Stanford University, Stanford, CA 94305, USA. [9]These authors contributed equally: Zhijian Li, Laura Amaya. ✉e-mail: wenderp@stanford.edu

T lymphocytes are key players in adaptive immune responses and a promising delivery target. They help to maintain immune homeostasis and provide essential immune protection against diverse viral, bacterial, and parasitic infections[16]. Effective modulation of T lymphocytes by gene delivery has been critical to the implementation of various immunotherapies, such as CAR-T cell therapy[17], immune checkpoint blockade[18], and in vivo T-cell reprogramming[19]. However, gene delivery to primary T lymphocytes is notoriously difficult. Electroporation is commonly used to transfect T lymphocytes with mRNA[20], but it is challenging to scale up, incurs cell damage and is unsuitable for most in vivo applications. Viral vectors have also been investigated for T-lymphocyte transduction[21] but their use is often limited by immunogenicity and gene cargo size. To circumvent these problems, considerable effort has been made to optimize LNP systems for T lymphocyte transfection. One strategy uses LNPs (e.g., Dlin-MC3-DMA) conjugated to T-cell targeting antibodies[22–24] such as anti-CD3, -CD4, and -CD5 to target T cells in vivo (Fig. 1a). While up to 75% T-cell transfection is observed in lymphoid organs such as the spleen, most transfections still favor LNP delivery to the liver. Moreover, T-cell depletion has been observed after treatment of anti-CD3 conjugated LNPs[22], raising potential safety concerns for the implementation of this strategy. In important recent advances, chemical structures of LNPs have also been systematically tuned for T-cell transfection without targeting ligands (e.g., 93–017 S)[25,26]. However, this approach currently requires high doses of mRNA and repeated injections to achieve suitable cell transfection, underscoring the continued need for more effective delivery methods targeting T lymphocytes.

Polymers have provided the structural foundation for new classes of mRNA delivery systems and many show considerable potential for efficient T-cell transfection. For example, a comb-shaped polymer (CP-25-16) has been developed which transfects 25% of primary human T cells in vitro[27] (Fig. 1a). In addition, a poly beta-amino ester (PBAE-447) system formulated with anti-CD3 is reported to transfect 80% of T cells in vitro and 8% of splenic T cells in vivo[28]. The Chang, Levy, Waymouth and Wender groups have also reported the use of charge-altering releasable transporters (CARTs) for delivery of polyanionic cargoes including mRNA, plasmid DNA, circRNA, and their combinations[5,28–30]. Prepared by organocatalytic ring-opening polymerization (OROP), CARTs are block oligomers consisting of an initiator, one or more lipid blocks and a polycationic block (Fig. 1b). The unique CART cationic block serves to electrostatically complex polyanions at acidic pH (<5.5), but at physiological pH (-7.4) the CART delivery system undergoes an oxygen-to-nitrogen (O-to-N) acyl shift[29], resulting in its irreversible conversion to a neutral lactam which triggers release of the polyanionic cargo[30].

Using a combinatorial lipid screening strategy, we previously identified CARTs incorporating both oleyl and nonenyl lipids (termed Oleyl-Nonenyl Amino CARTs, i.e., ONA CARTs) that showed significantly

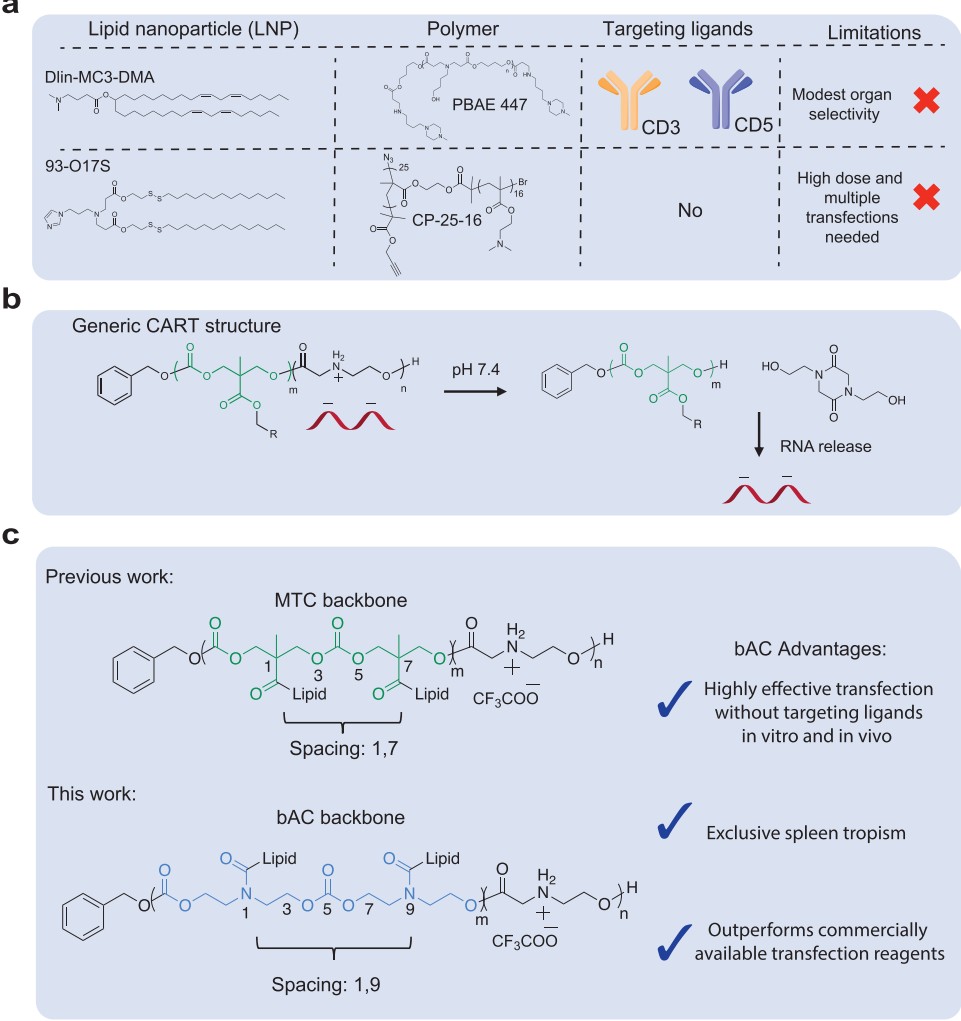

**Fig. 1 | Summary of nonviral delivery systems for primary T lymphocyte delivery. a** Representative nonviral delivery systems for primary T-cell transfection. **b** CART/mRNA complexes and their pH-driven nitrogen-to-oxygen acyl shift triggering charge cancellation and mRNA release. **c** bAC CARTs possess a polymeric backbone with distinct lipid spacing, leading to improved T-cell delivery.

improved (up to ~80%) transfection efficiency in Jurkat cells, a T lymphocyte cell line[31] (Fig. 1c). ONA CARTs have since been employed successfully in the clinical development of prophylactic (COVID) and therapeutic (cancer and metastatic disease) vaccinations[32,33] and in the generation of CAR-NK cells[32,33]. Despite their encouraging performance (80 %) in Jurkat cell transfection, ONA CARTs were less effective (10–20%) in the transfection of primary T cells. To address this problem, structural modifications of the CART cationic blocks and initiators have been investigated. Notably, we recently reported that CARTs employing fingolimod as the initiator[34], a small molecule targeting sphingosine-1-phosphate receptor (S1P1) on T and B lymphocytes, exhibited improved Jurkat cell transfection in vitro. However, transfection of T cells in the murine model was not improved.

To improve transfection efficiency, cell and organ selectivity and tolerability, CART structural studies have largely focused on variations in the lipid[35], initiator[30], and cationic subunits[35]. Much less attention has been directed at the CART polymeric backbone (main chain) and side chain spacing which would be expected to influence CART stability, hydrophobicity, flexibility, cargo release rates, and cell and organ selectivity. For example, in our studies on oligo-arginine transporters, we reported that insertion of non-consecutive aminocaproic acid spacers in the oligomeric backbone resulted in better cellular uptake than unspaced transporters[36]. More recently, we synthesized guanidinium-rich transporters functionalized with glycerol-derived polymeric backbones. Compared to methyl-trimethylene carbonate backbone (MTC) transporter analogs, glycerol backbone transporters improved nanoparticle stability when complexed with siRNA and allowed for control of siRNA release rates[37]. All of our previously reported CARTs, however, have been limited to the MTC backbone and its side chain spacing.

Here we introduce a CART delivery system with a beta-amido carbonate backbone (bAC), which we have found can be accessed using an amido-variant of previously reported 8-membered cyclic carbonates incorporating nitrogen as a tertiary amine or urethane[38]. Compared to the MTC backbone, the bAC backbone differs in composition and side chain spacing (Fig. 1c).

In this work, we investigated the delivery efficiency of the bAC backbone system. We accomplished this by synthesizing a library of 24 bAC CARTs with different lipid components using a step economical organocatalytic ring-opening oligomerization. The best-performing bAC CARTs showed up to 70% primary T-cell transfection in vitro, significantly improving on the performance of our lead ONA CART (<20%). Importantly, systemic in vivo delivery of the best bAC CARTs yielded up to 97% spleen tropism and transfected 8% of primary splenic T cells without T-cell targeting ligands. Moreover, a direct comparison of bAC-CART and MTC analogs with identical lipids and cationic structures indicated that the bAC backbone was the key contributor to enhanced protein expression in vitro and in vivo.

## Results
### Organocatalytic ring-opening synthesis of bAC CARTs
Many polymeric backbones derive from 6- or 7-membered cyclic ester and carbonate monomers due to the ease of monomer synthesis and the favorable thermodynamics for polymerization. Based in part on our studies on cell-penetrating guanidinium-rich spaced molecular transporters (peptides and peptoids)[39,40], we set out to determine whether cyclic monomers of larger ring sizes could significantly impact transporter properties by changing the polymer flexibility, backbone composition and spacing of attached side chains. Yang, Hedrick and colleagues recently reported the synthesis and polymerization of a series of 8-membered cyclic carbonate monomers incorporating nitrogen in the ring as a tertiary amine or urethane[41]. We set out to determine whether attachment of the corresponding beta-amido carbonate oligomer (bAC) as a lipid block to an oligo-alpha-amino ester as a cationic block would produce a new class of CARTs, bAC CARTs, with improved transfection efficacy and organ

selectivity. We first attempted to synthesize a bAC monomer functionalized with a lauroyl lipid linkage (bAC monomer 1, Fig. 2a). The lauroyl functionalized diol (Lauroyl DEA) was obtained in >99% yield by coupling lauroyl chloride with diethanolamine. Based on a related procedure, synthesis of bAC monomer 1 with ethyl chloroformate or triphosgene proceeded in only 25-30% yield[39,40,42]. After screening various cyclization reagents and conditions, we found that slow triphosgene addition to diluted diol reactants at −20 °C improved the yield to 45–51%. The resulting bAC-1 was then copolymerized with Boc-protected morpholinone using our previously reported organocatalytic ring-opening polymerization (OROP) methodology with benzyl alcohol as the initiator[39] (Fig. 2b). After TFA deprotection to remove the Boc groups, the resulting bAC CARTs (bAC-1a) incorporated a block of ~17 lipid units and a block of ~10 cationic units.

### bAC CARTs mediate mRNA delivery into Jurkat cells with enhanced transfection efficacy
We complexed bAC CARTs with 200 ng of eGFP mRNA by simple mixing at pH 5.5 and used the resulting bAC CART complexes to transfect Jurkat cells (200 ng mRNA/100,000 cells). We evaluated transfection performance by flow cytometry 24 h after transfection (Fig. 2c). A CART with a dodecyl lipid called MTC-1a (MTC analog of bAC-1a) and our most efficient previously reported CART for Jurkat cell transfection (ONA) were included for comparative analysis. The use of bAC-1a resulted in >70% eGFP transfection of cells, significantly outperforming both MTC-1a (8%) and ONA CART (46%). Encouraged by this result, we synthesized seven additional bAC monomers (2–8) with different lipid attachments (Fig. 2d). In addition to previously reported straight chain and unsaturated lipids (bAC monomers 1–3), we included an isoprenoid lipid (bAC monomer 4) based on its improved performance in MTC CARTs. We also included four branched lipids derived from an MTC scaffold (bAC monomer 5–8). These were included to probe lipid size effects and further motivated by recent LNP literature showing enhanced transfection and biodegradability by incorporating branched lipids with ester linkages[30]. Eight bAC monomers were prepared each with a different lipid and each was oligomerized as described above (polymerization conditions described in Supplementary Table 1) with a Boc-protected morpholinone monomer to produce three lipidated bAC CARTs differing in the lengths of the lipid and cation blocks as determined by 1H-NMR. Specifically, block length "a" CARTs contained ~17 lipids and ~10 cations, block length "b" CARTs contained ~10 lipids and ~10 cations, and block length "c" CARTs contained ~10 lipids and ~17 cations, giving a total of 24 bAC CARTs to evaluate the effects of both lipid type and lipid-to-cation ratios on the performance of the bAC CART system.

To allow head-to-head comparison of bAC and MTC CARTs for mRNA delivery, we additionally synthesized four MTC monomers, each of which was subsequently polymerized to provide the corresponding MTC-1b, MTC-4b, MTC-7b, and MTC-8b CARTs (Fig. 2e). The polydispersity of the synthesized bAC CARTs and MTC CARTs were evaluated by gel permeation chromatography (Supplementary Table 1) and ranged from 1.14 to 1.37. The size of nanoparticles formed by CART/mRNA complexes was evaluated by dynamic light scattering (DLS). bAC CART/mRNA complexes formed particles ranging in size from 130 to 280 nm (Fig. 2f), with no apparent trend across lipid types or block lengths. The bAC backbone did not have a major effect on CART/mRNA complex size (Fig. 2g). The bAC CARTs were subsequently evaluated for Jurkat cell transfection as described above but using a lower mRNA dose (100 ng). Top performers bAC-4b, bAC-4c, and bAC-7c were identified and reached a maximum of 78% eGFP+ cells (100 ng mRNA/100,000 cells) (Fig. 2h). The transfection performance of each CART was also evaluated by integrating fluorescence intensity of all eGFP+ cells, here termed Area Under the Curve or AUC (Fig. 2i), which indicated the strength of protein expression. Based on this AUC comparison, the bAC CARTs outperformed the ONA CART in

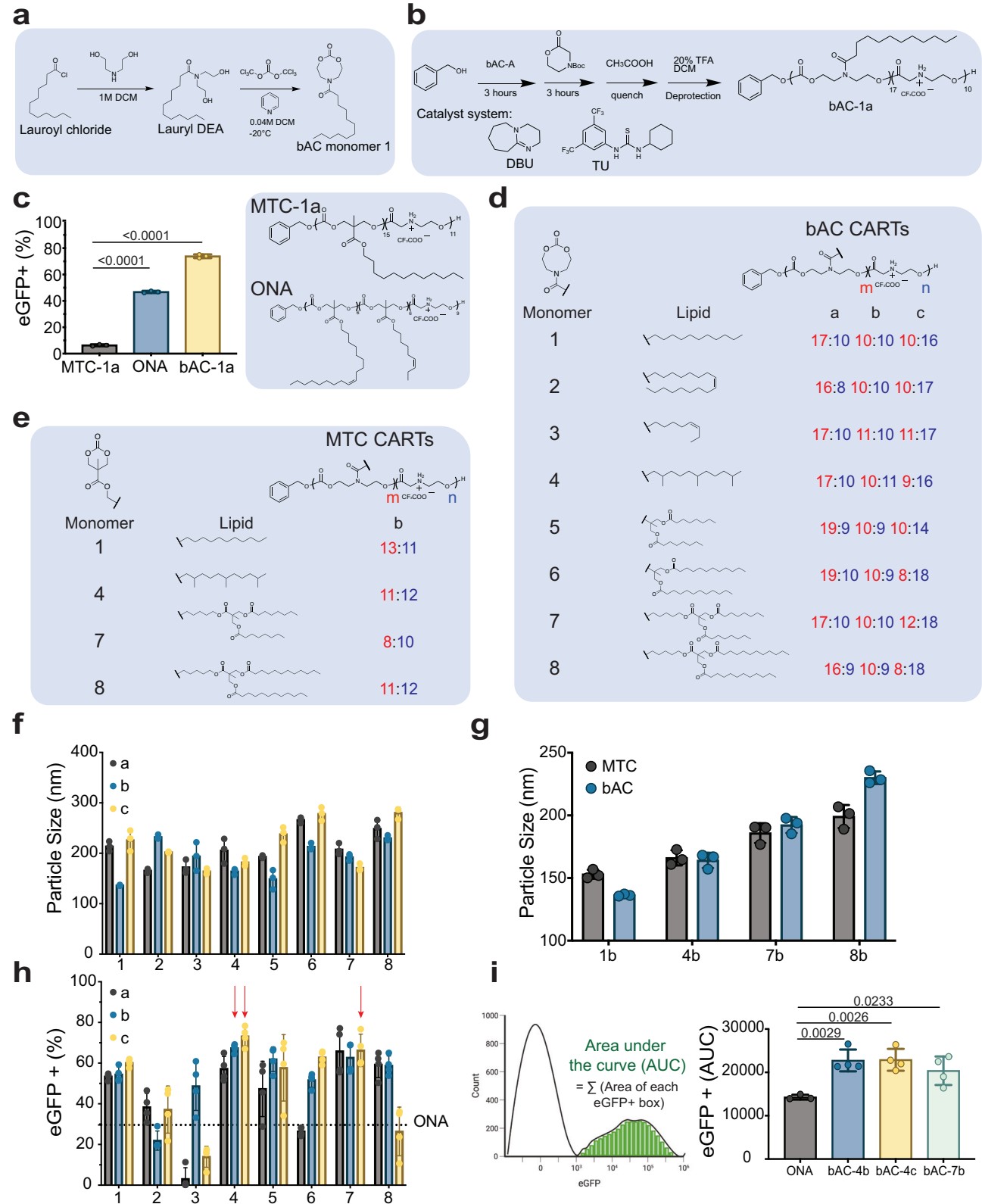

transfecting Jurkat cells (Fig. 2i). Minimal toxicity was observed for all bAC CARTs during the transfection studies (Supplementary Fig. 1).

## bAC polymeric backbone improves primary T lymphocyte transfection in vitro

While the Jurkat cell line is commonly used as a proxy to evaluate T-cell transfection efficacy due to its ease of access and maintenance,

it is still significantly different from primary T lymphocytes[2,43], which are more relevant to clinical applications. For example, we previously reported up to 80% transfection with ONA CART and eGFP mRNA (200 ng mRNA/25,000 cells) in Jurkat cells, but <20% of primary T lymphocytes were transfected under the same conditions. Others have similarly reported reduced transfection of primary T lymphocytes compared to Jurkat cells[44]. To address this issue, we evaluated

**Fig. 2 | bAC CARTs mediate highly efficacious mRNA delivery to Jurkat cells.**
**a** Schematic representation of bAC monomer synthesis. **b** Polymerization scheme of bAC CARTs with TU/DBU catalyst system (see Methods section). **c** Jurkat transfection with eGFP mRNA (200 ng mRNA/100,000 cells) comparing MTC-1a, ONA, and bAC-1a CARTs. ($n = 3$, bars represent mean values +/− SD). Statistical significance was calculated using two-way ANOVA. **d** Chemical structures of bAC CART library. **e** Chemical structures of MTC CART analogs. **d**, **e** The numbers in red indicate lipidic block length, represented by the subindex "m", and numbers in blue indicate cationic block length, and represented by the subindex "n". **f** Particle size

of bAC CART/mRNA complexes ($n = 3$, bars represent mean values +/− SD) and **g** Particle size comparison to MTC CART/mRNA complexes ($n = 3$, bars represent mean values +/− SD). **h** eGFP expression of bAC CARTs in Jurkat cells (100 ng mRNA/100,000 cells). Dotted line indicates ONA eGFP expression level as baseline, and red arrows indicate the top three performers ($n = 4$, bars represent mean values +/− SD). **i** Area under the curve (AUC) computation of eGFP expression (left) and eGFP+ AUC (right) levels among the best CARTs ($n = 4$, bars represent mean values +/− SD). Statistical significance was calculated using one-way ANOVA with Tukey's multiple comparisons test.

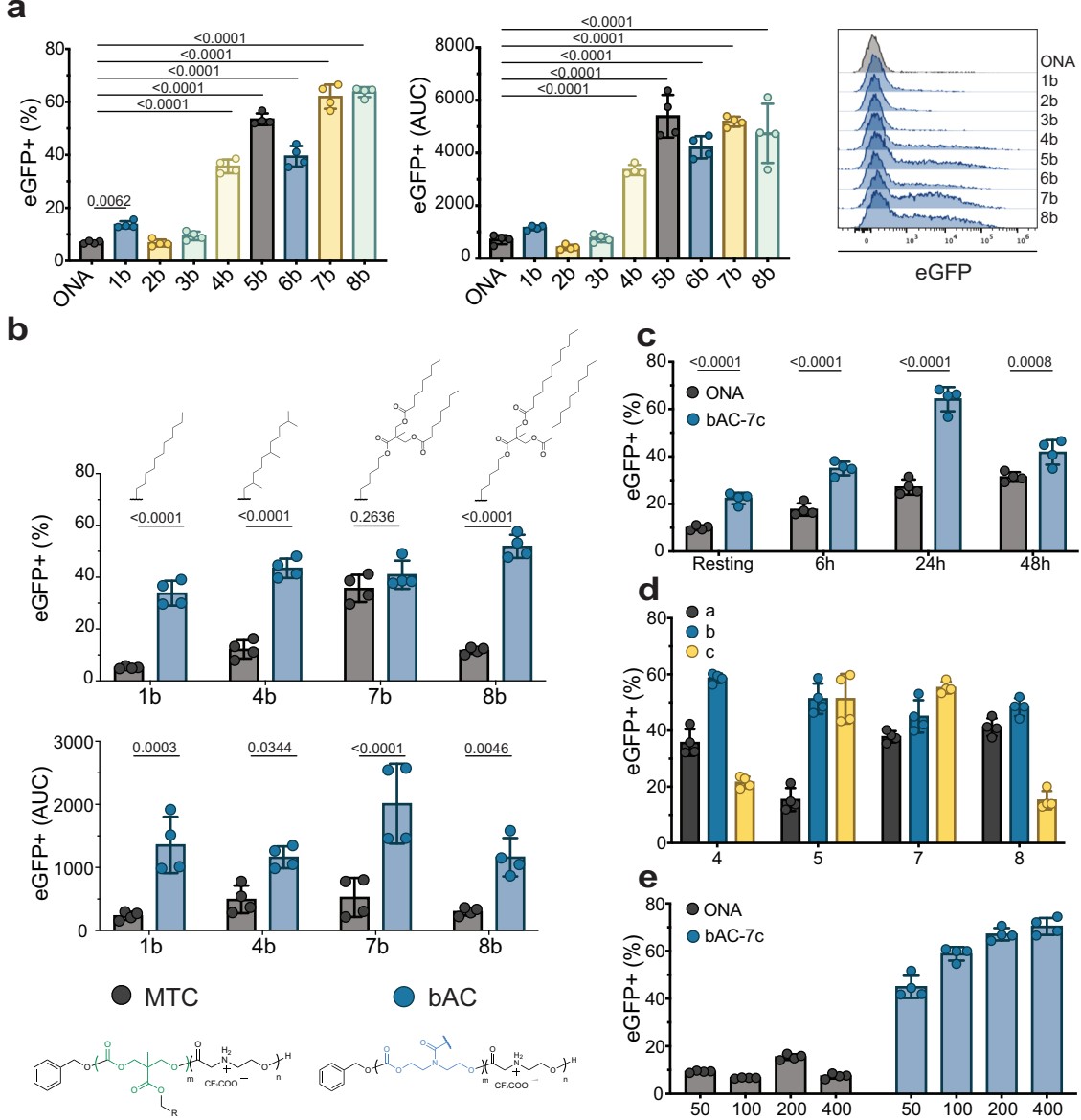

**Fig. 3 | bAC polymeric backbone improves primary T lymphocyte transfection in vitro. a** bAC CART screening in primary human T cells. Percentage of eGFP+ cells (left) and AUC fluorescent signal (right) 24 h after transfection. ($n = 4$, bars represent mean values +/− SD). Statistical significance was calculated using one-way ANOVA with Tukey's multiple comparisons test. **b** Percentage of eGFP+ cells (top) and AUC fluorescent signal (bottom) 24 h after transfection of CARTs with bAC backbone versus MTC backbone. ($n = 4$, bars represent mean values +/− SD).

Statistical significance was calculated using two-way ANOVA. **c** Effect of activation time on T-cell transfection with CARTs. ($n = 4$, bars represent mean values +/− SD). Statistical significance was calculated using two-way ANOVA. **d** Effect of lipid length screen on T-cell transfection with bAC CARTs ($n = 4$, bars represent mean values +/− SD). **e** Effect of mRNA dose (in ng) on CD8 + T cells transfection with CARTs. ($n = 4$, bars represent mean values +/− SD).

bAC CARTs 1b-8b for delivery of eGFP into human T lymphocytes freshly isolated from peripheral blood mononuclear cells (PBMC) (Fig. 3a). Compared to ONA CART, which transfected only ~10% of primary T cells, we observed up to 65% transfected cells with bAC-7b

and >40% transfection with several other bAC CARTs (bAC-5b, −6b, −8b), indicating that bAC CARTs transfect more primary T cells than the ONA CART complexes. In addition, bAC-7 resulted in a fivefold increase of AUC over ONA, indicating the amount of protein

expression was also increased. All CARTs showed a slight preference for CD8+ T cell transfection (Supplementary Fig. 2). To determine if the improved delivery was influenced by lipid type or the polymeric backbone and lipid spacing, we compared the transfection efficiency of bAC-1b, −4b, −7b, and −8b with MTC-1b, −4b, −7b, and −8b in primary T lymphocytes (Fig. 3b). bAC-1b, −4b, −8b significantly outperformed their corresponding MTC analogs in terms of percentage of eGFP+ cells (e.g., bAC-1b vs. MTC-1b). All bAC CARTs also outperformed all MTC analogs in AUC, suggesting that the bAC backbone and side chain spacing are crucial for improving primary T-cell transfection, regardless of lipid identity.

We next sought to further optimize primary T lymphocyte transfection with the bAC CARTs. Activation is typically required for efficient T-lymphocyte transfection, but activation methods (choice of ligands or beads type) and time of cell activation could significantly impact delivery efficacy. We tested the effect of different activation times on CART delivery and protein translation and found the efficacy of mRNA delivery to be time-dependent for both ONA and bAC CARTs (Fig. 3c). bAC CARTs achieved a peak of expression after 24 hours of activation (65% transfection rate) and significantly outperformed ONA at every time point. Interestingly, bAC CARTs also transfected >25% of non-activated T cells (Fig. 3c), which could be useful for research when cell activation needs to be avoided. With the optimal activation procedure, we next evaluated the delivery efficacy of bAC CARTs with the four most potent lipids (Lipids 4, 5, 7, 8) with varying block lengths (a, b, c) (Fig. 3d). All bAC CARTs with block length "b" demonstrated efficient T cell transfection, while block length "a" analogs overall showed comparatively decreased transfection. Three selected bAC CARTs (bAC-4b, bAC-5b, and bAC-7c) were further optimized by varying their charge ratios (CART Nitrogen to RNA Phosphate, N:P). A 10:1 charge ratio for bAC CARTs led to the highest rate of transfection (Supplementary Fig. 3). The best-performing CART, bAC-7c, was then evaluated with various mRNA doses. 50% transfection of primary T cells was achieved with as little as 50 ng of mRNA and a transfection plateau of 70% was realized with 400 ng of mRNA (Fig. 3e). No cell toxicity was observed at any dose (Supplementary Fig. 4). To assess the state of activation and functionality of T cells after CART transfection we measured the levels of activation markers CD25 and CD69, and exhaustion marker PD1. Our results show no difference in phenotype and activation state between CART transfected T cells and untreated controls (Supplementary Fig. 5a). We measured the capacity of transfected T cells to produce effector cytokines IFN-γ and TNF-α after 24 h stimulation with Dynabeads Human T-Activator CD3/CD28 and observed no significant differences between bAC-7c and untreated samples (Supplementary Fig. 5b). Finally, we measured the T cell proliferation potential after non-specific activation with anti-CD3/CD28 and the cytokine IL-2 and found no significant differences in cell growth and viability after CART transfection (Supplementary Fig. 5c). These data show the maintenance of phenotype and functionality of human T cells after transfection with bAC-7c CART.

To evaluate the performance of bAC CARTs relative to existing delivery methods, we compared bAC-7b with commercially available transfection reagents (Supplementary Fig. 6a and electroporation (Fig. 4a). We observed higher protein expression based on the eGFP+ AUC fluorescent signal when mRNA was delivered with bAC CART compared to all other transfection methods. All bAC delivery systems maintained good cell viability after treatment (Supplementary Fig. 6b, c).

In addition to delivery performance, the stability of bAC CART/ mRNA formulation is also critical for broader clinical applications. While beyond the scope of this initial study, the addition of 20% sucrose as a cryoprotectant to bAC CART/mRNA complexes allowed for their storage at −80 °C for at least one week without loss of delivery efficacy (Supplementary Fig. 7).

## bAC CARTs delivery enables ex vivo generation of highly cytotoxic CAR-T cells

To demonstrate the therapeutic potential of bAC CARTs, we transfected activated CD8+ T cells with anti-human CD19 (hCD19) mRNA and assessed the function of the resulting anti-hCD19 CAR-T cells by co-culturing with Nalm6-GL, a B-cell precursor leukemia cell line that stably express GFP and firefly luciferase (Fig. 4b). Twenty hours after transfection we observed significantly higher levels of anti-hCD19 CAR expression using bAC-7c compared to ONA CART (Fig. 4c and Supplementary Fig. 8a). To assess the impact of anti-hCD19 CAR on CD8+ T cell function, we analyzed the levels of CD107a, an indicator of degranulation activity, as well as the expression of cytokines IFN-γ and TNF-α in transfected and untransfected CD8+ T cells during co-culture with Nalm6-GL cells at an E:T ratio of 1:4. Compared to untransfected CD8+ T cells and cells transfected with mCherry (irrelevant) mRNA, CD8+ T cells transfected with anti-hCD19 mRNA and both ONA or bAC-7c showed a higher frequency of CD107a expression (as a marker of degranulation) and an increased frequency of IFN-γ and TNF-α expression (Fig. 4d and Supplementary Fig. 8b), suggesting elevated activation and function due to CAR expression. In addition, we co-cultured wild-type or CD19 knock-out Nalm6-GL cells with CAR-T cells transfected with ONA/hCD19 or bAC-7c/hCD19 at 10:1 effector-to-target (E:T) ratio, and demonstrated that anti-hCD19 CAR-T cells had significantly increased killing of Nalm6-GL cells after 13 h compared to untransfected cells or cells transfected with mCherry mRNA (Fig. 4e and Supplementary Fig. 8c). In contrast, co-culturing with anti-hCD19 CAR-T cells did not lead to increased killing of CD19 knock-out Nalm6-GL cells (Fig. 4e), indicating the dependence of Nalm6-GL cell killing by anti-hCD19 CAR-T cells on CD19 expression. Overall, these findings demonstrate that bAC-7c transfection enables high expression of anti-hCD19 CAR in CD8+ T cells, leading to specific targeting of Nalm6 leukemia cells through CD19-CAR interaction.

## bAC CARTs induce a faster change in nanoparticle surface charge and earlier protein expression

We next sought to probe what factors contributed to the better T lymphocyte delivery efficacy of bAC CARTs compared to MTC CARTs. One distinct characteristic of the CART system is its ability to transform from polycationic oligomers to neutral products through a charge alteration process (O-to-N acyl shift) (Fig. 5a). This transformation is monitored by measuring the change of surface charge (zeta potential) of CART/mRNA complexes. Compared to ONA and MTC-7b CART, bAC-7b CART exhibited a significantly faster decrease in zeta potential (Fig. 5a, b), which correlated with the transfection efficacy of the three CARTs (bAC-7b > MTC-7b > ONA). Since the charge alteration process reduces available cationic charges binding to polyanionic mRNA, we hypothesized that bAC-7b could lead to faster mRNA release due to its faster charge neutralization. To investigate this, we developed an assay to measure RNA release with a commercial dye (Qubit RNA HS assay kit). In essence, the Qubit dye induces fluorescence upon binding to free mRNA, which is blocked by CART/mRNA complexation (Fig. 5c). As CARTs degrade, more mRNA becomes accessible and the release kinetics could be monitored by measuring the increase in fluorescence. We observed <4% fluorescence signal compared to control with free mRNA for all three CARTs immediately after formulation, suggesting >96% mRNA was encapsulated. Surprisingly, MTC-7b had the fastest mRNA release rate (Fig. 5d), followed by bAC-7b which released mRNA slightly faster than ONA CART. This indicates that the kinetics of surface charge change differs from the kinetics of mRNA release. This is not unreasonable as charge change involves a single N-to-O acyl shift whereas release requires multiple shifts. In addition, the surface and internal microenvironments differ. This could potentially explain why bAC-7b has faster surface charge change but slower mRNA release kinetics. This difference in the kinetics of surface and internal processes is not unexpected as lipid type and side chain spacing could alter

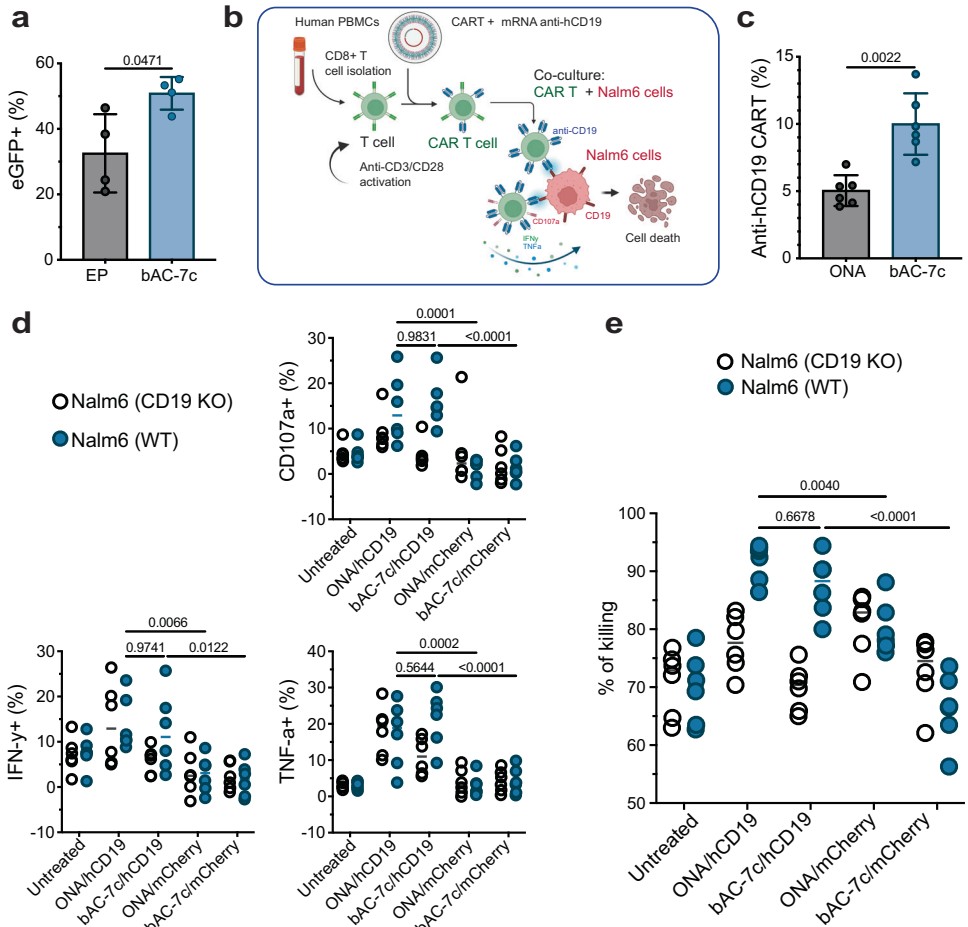

**Fig. 4 | bAC CARTs delivery enables ex vivo generation of highly cytotoxic CAR-T cells. a** CART delivery of mRNA with 10:1 charge ratio compared to electro-poration (EP) of naked mRNA previously optimized for primary T-cell delivery based on the manufacturer's recommendations (see Methods section) (*n* = 4, bars represent mean values +/− SD). Statistical significance was calculated using unpaired two-tailed *t* test with Welch's correction. **b** Schematic representation of CD8+ T cells transfected with ONA and bAC-7c CARTs complexed with anti-human CD19-CAR mRNA and co-culture assay with Nalm6-GL cells (labeled as Nalm6 in the figure) cells to assess antigen-specific killing and functional phenotype. Created with BioRender.com. **c** Percentage of anti-hCD19 expression 20 h post-transfection of anti-hCD19 mRNA with ONA and bAC-7c (*n* = 6, bars represent mean values +/−

SD). Statistical significance was calculated using a two-tailed Mann–Whitney test. **d** Percentage of cells expressing degranulation marker CD107a and activation markers IFN-γ and TNF-α after co-culture with Nalm6-GL cells at 1:4 effector: target (E:T) ratio. The data in the groups treated with CART/mRNA complex was nor-malized by the baseline marker expressions of untransfected T cells in the same co-culture well, and the raw data before normalization can be found in Supplementary Fig. 8d. (*n* = 6, bars represent median). Statistical significance was calculated using two-way ANOVA. **e** Percent of cell killing (Nalm6-GL cells) after co-culture with anti-hCD19 expressing CAR-T cells at 10:1 effector: target (E:T) ratio (*n* = 6, bars repre-sent median). Statistical significance was calculated using two-way ANOVA.

particle packing, particle viscosity, and proton exchange differentially in different particle domains.

Several studies have suggested that mRNA uptake is positively correlated with protein expression in T lymphocyte after transfection[27]. To assess if bAC-7b might enable greater mRNA uptake, we complexed CARTs with a 1:1 ratio of Cy5-labeled non-translatable mRNA and non-labeled eGFP mRNA and delivered them into primary T lymphocytes, which allowed us to independently measure mRNA uptake (Cy5 + ) and protein translation (eGFP +). Interestingly, there was no significant difference in mRNA uptake between MTC-7b and bAC-7b CARTs in primary T cells at different timepoints, even though both were higher than ONA (Fig. 5e). This suggests that improved mRNA uptake is not the cause of delivery enhancement for bAC CARTs. Notably, MTC-7b had higher Cy5+ mRNA uptake than bAC-7b in Jurkat cells (Supple-mentary Fig. 9a), highlighting a difference of Jurkat and primary T cells, and the limitations of using Jurkat cells as a model to evaluate T cell transfection reagents. When evaluating eGFP protein expression at different timepoints, we found that bAC-7b delivery induced sig-nificantly faster translation kinetics, resulting in eGFP+ cells as early as

30 min after transfection, a time at which MTC-7b and ONA CART delivery showed no eGFP expression (Fig. 5e and Supplementary Fig. 9b). These results suggested that bAC CARTs induce faster change of surface charge compared to MTC CARTs and result in earlier onset of protein expression (i.e., the time between administration and detectable reporter signal).

To gain further insights into the potential reasons for the earlier onset of protein expression in bAC CARTs, we employed confocal microscopy to track Cy3-labeled mRNA delivered by CARTs. The results were intriguing, revealing that bAC-7c exhibited notably more localized mRNA distribution compared to ONA CART and MTC-7c in Jurkat cells (Supplementary Fig. 10). This correlation aligns with the observed trend in protein expression onset. The detailed mechanism behind this phenomenon is currently under active investigation.

### bAC CARTs efficiently transfect T lymphocytes in vivo with good tolerability
Encouraged by their efficient delivery to primary T lymphocytes in vitro, we sought to determine if bAC CARTs could transfect T cells

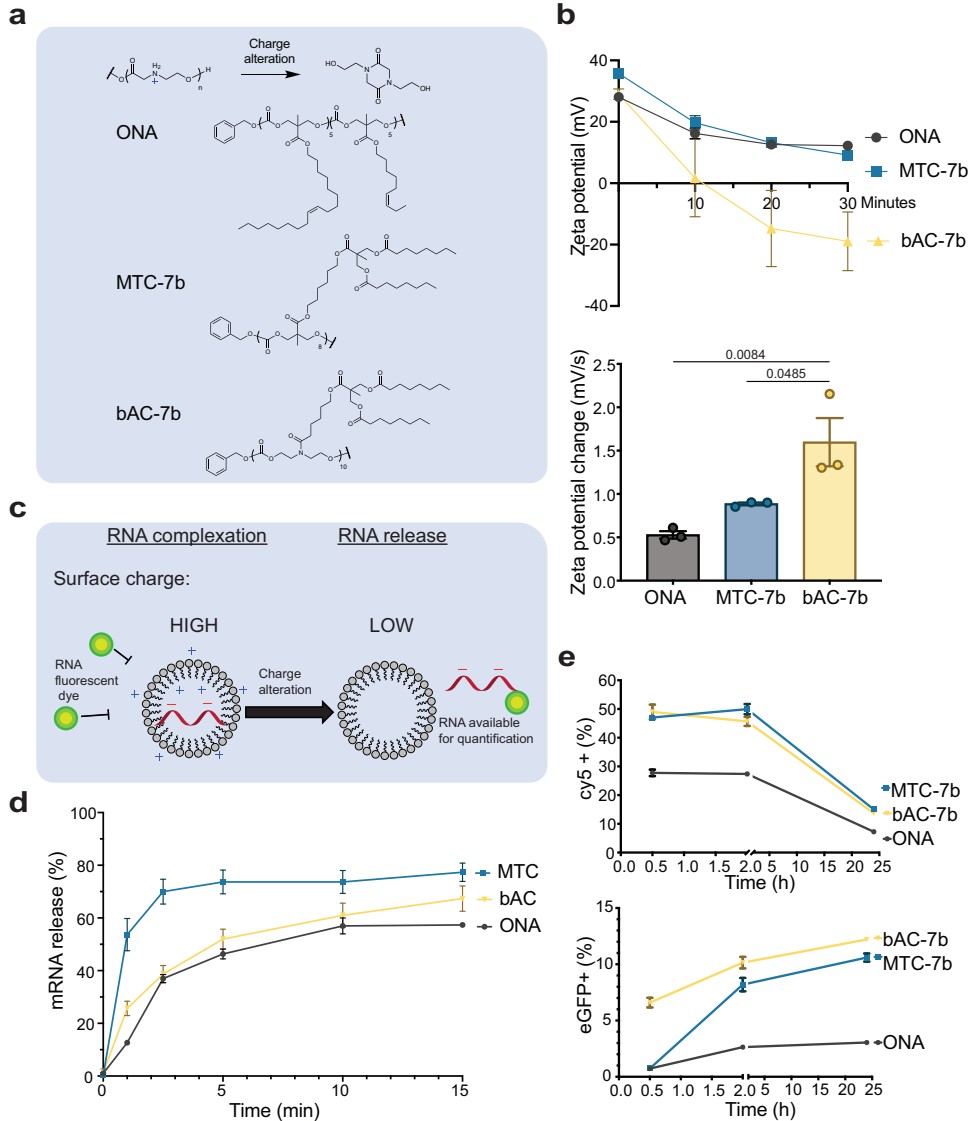

**Fig. 5 | bAC CARTs induce faster change in nanoparticle surface charge and earlier protein expression. a** Structures of CARTs included in this study. **b** Kinetics of surface charge change upon CART/mRNA complexation (top, $n = 3$, bars represent mean values +/− SEM). The bar graph on the bottom represents the average rate of surface charge change from 0–30 min. ($n = 3$, bars represent mean values +/− SEM). Statistical significance was calculated using one-way ANOVA with Tukey's multiple comparisons test. **c** Schematic representation of methods to monitor mRNA release from CART/mRNA nanoparticles. Created with ChemDraw. **d** Kinetics of mRNA release upon CART/mRNA complexation ($n = 3$, bars represent mean values +/− SEM). **e** Kinetics of Cy5+ labeled mRNA uptake (top) and eGFP mRNA protein expression (bottom) in primary T cells with CART delivery ($n = 4$, bars represent mean values +/− SEM).

in vivo. We first complexed six bAC CARTs (bAC 4a-c, 7a-c) with luciferase mRNA and delivered each complex intravenously (i.v.) to BALB/c mice (Fig. 6a, b). After administration of D-luciferin, luciferase protein expression was evaluated 6 h post-injection by whole-body imaging with a charged-coupled device camera. bAC-4b had a stronger luciferase signal than bAC-4a and −4c, while bAC-7c outperformed bAC-7a and −7b, all consistent with results from the in vitro transfection of primary T cells. However, luciferase expression with bAC-4b delivery was 2.5-fold higher than that with bAC-7c, even though bAC-7c slightly outperformed bAC-4b in vitro. The effect of charge ratios was evaluated for bAC-4b and bAC-7c, and a 10:1 charge ratio was found to be optimal for delivery (Supplementary Fig. 11). We then investigated if the bAC backbone could improve in vivo delivery by comparing bAC-4b with MTC-4b and bAC-7b with MTC-7b (Fig. 6c). Encouragingly, both bAC-4b and bAC-7b significantly outperformed their MTC analogs, exhibiting up to tenfold more luciferase expression.

To evaluate payload biodistribution following bAC CART delivery, we isolated spleens, lungs, and livers and quantified their luciferase activity (Fig. 6d, e). Both bAC-4b and bAC-7c showed excellent selectivity for spleen uptake (>90 %), comparable to ONA CART as previously reported[31,45]. Importantly, bAC-4b led to splenic luciferase activity threefold higher than ONA CART, suggesting that bAC-4b is a more efficient mRNA delivery system in vivo.

To address the cell specificity of CART complexes when delivered in vivo, we used a Cre recombinase murine model (Ai14 mice)[33] to measure specific cell recombination after i.v. delivery of mRNA Cre complexed with different CARTs (Fig. 7a). With this mouse model, we can detect the cells that internalize mRNA and efficiently translate the Cre recombinase protein as they will express the fluorescent protein tdTomato following Cre-mediated recombination. After i.v. delivery of 15 μg of mRNA Cre in Ai14 mice, we observed Cre-mediated recombination (tdTomato +) in most subsets of CD45+ leukocytes in the spleen, including T cells (CD4+ and CD8 +) and antigen-presenting

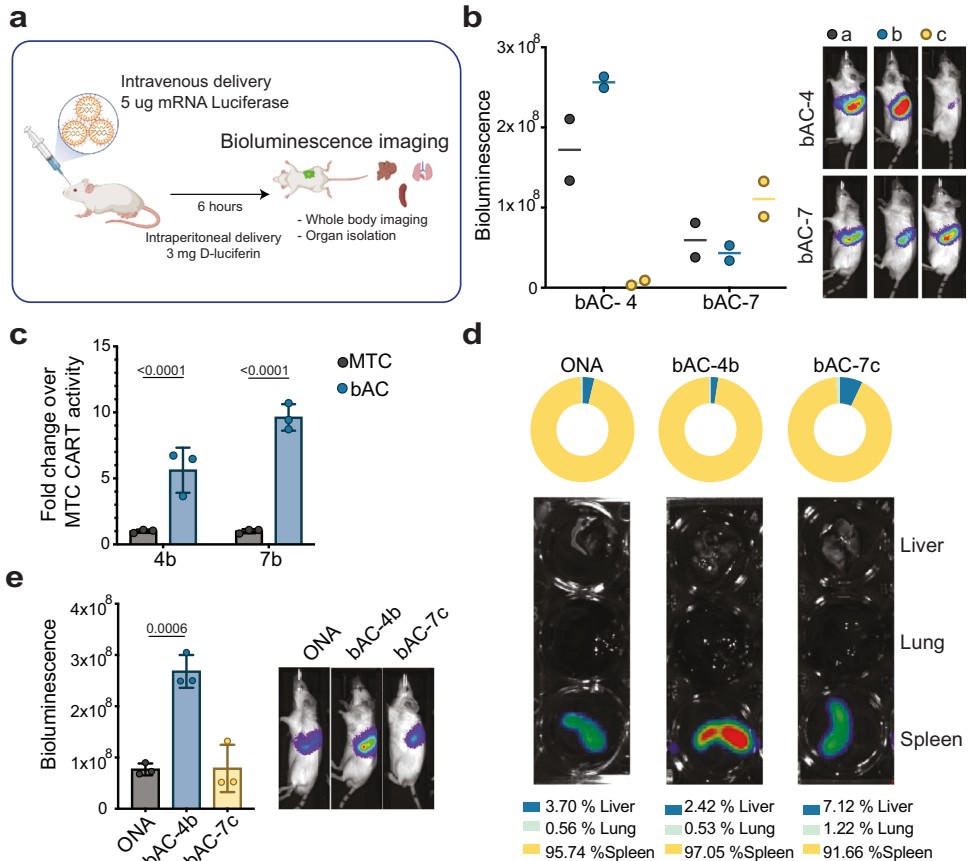

**Fig. 6 | bAC CARTs efficiently transfect T lymphocytes in vivo. a** Schematic representation of mRNA delivery in vivo to assess biodistribution. Created with BioRender.com. Briefly, 5 µg of Luciferase mRNA were complexed with each CART and delivered retro-orbitally. Bioluminescence imaging was acquired 6 h after transfection. **b** Effect of bAC CART's block length (a, b, c) on luciferase mRNA delivery efficiency ($n = 2$, bars represent median). **c** bAC CART delivery of luciferase mRNA outperforms MTC CART analogs ($n = 3$, bars represent mean values +/− SD). MTC CART activity was normalized to onefold. Statistical significance was calculated using two-way ANOVA. **d** Biodistribution of luciferase protein expression after i.v. mRNA delivery with selected CARTs. **e** Quantitation of luminescence signal in spleen after i.v. delivery with selected CARTs ($n = 3$, bars represent mean values +/− SD). Statistical significance was calculated using a two-tailed unpaired $t$ test.

cells such as dendritic cells (CD11c + ), macrophages (F4/80 + ), and B cells (CD19 + ) (Supplementary Fig. 12). As expected, most of the recombined cells were B cells (40-50%), which represent >55% of the cells in the mouse spleen. No significant difference in cell proportions was observed between CARTs (Fig. 7b). Importantly, we achieved up to 8% Cre-mediated recombination in CD4+ and CD8+ T cells in the spleen with bAC-7c CART without any targeting ligands. Interestingly, bAC-7c resulted in a twofold increase in T-cell transfection compared to bAC-4b (Fig. 7c), even though bAC-4b resulted in higher luciferase activity.

As the A14 model only provides information on the percentage of cells transfected but not on protein expression levels, we measured protein translation directly on T cells after in vivo delivery of mRNA with bAC or ONA CARTs. We injected luciferase mRNA and isolated T cells 6 h after injection and measured luciferase expression in a plate format (Fig. 7d). Delivery of mRNA luciferase with bAC CART demonstrated a threefold increase in luciferase expression in splenic T cells compared to ONA CART (Fig. 7e). These data indicates that bAC-7c induces more protein expression in splenic T cells than ONA CART. We further tested different doses (2, 5, 10 µg) of luciferase mRNA and observed dose-dependent luciferase activity in splenic T cells, indicating transfection could be boosted with a higher mRNA dose (Fig. 7f). These doses are comparable to those used clinically (e.g., 6 µg mRNA for a 20 g mouse, corresponding to 0.3 mg/kg).

To determine the intra-spleen biodistribution of mRNA after delivery with bAC CART we used mRNA covalently labeled with Cy5. Briefly, 2 h after i.v delivery of Cy5-mRNA with bAC-7c, spleens were isolated and the levels of Cy5-mRNA in splenocytes was quantified by flow cytometry. We showed that on average 85% of Cy5-mRNA is internalized by CD45+ splenocytes (Supplementary Fig. 13a), which is equally distributed among dendritic cells, macrophages, and B cells (20%). Only a small proportion is localized to CD8 + T cells (5%) and CD4 + T cells (1%) (Supplementary Fig. 13b).

Assessing in vivo tolerability is a critical aspect in determining the clinical viability of drug delivery systems. Hence, we conducted a comprehensive preclinical toxicology blood analysis following the administration of bAC CART/mRNA formulations. For comparison, we formulated FDA-approved MC3 LNP components with mRNA luciferase and utilized them as a benchmark[46]. When compared to MC3 formulations, bAC formulations exhibited similar levels of blood metabolites, which serve as indicators of liver (ALT, AST, ALP) or kidney (blood urea nitrogen, calcium, and phosphorus) damage (Supplementary Fig. 14a). These findings suggest the absence of clinical signs of acute toxicity or alterations in clinical pathology. Furthermore, we examined whether bAC CART delivery prompted any dysregulation in blood counts or undesirable inflammatory responses post-administration. Blood counts remained normal and were comparable to the MC3 formulation (Supplementary Fig 14b). Similarly, the levels of inflammatory cytokines, including IL-1b, IL-6, and TNFa, remained

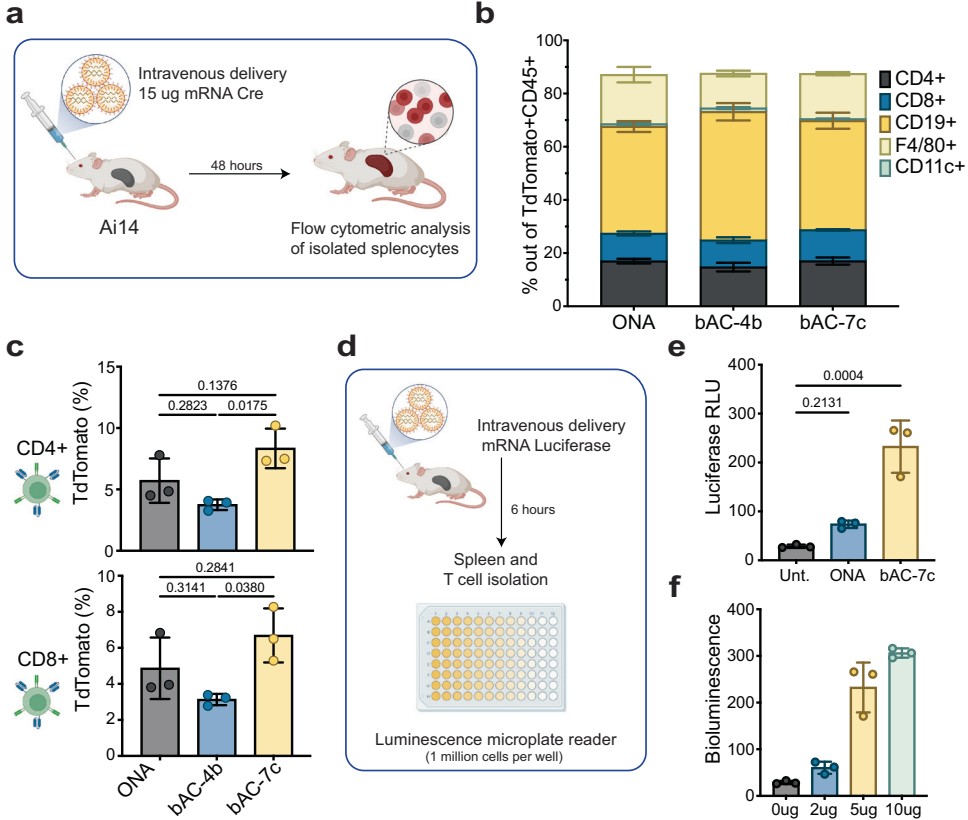

**Fig. 7 | Cell specificity after i.v. delivery of bAC CARTs. a** Schematic representation of mRNA delivery in vivo to assess cell specificity. Created with BioRender.com. Briefly, 15 μg of Cre mRNA were complexed with each CART and delivered retro-orbitally. Flow cytometric analysis of isolated spleens was performed 48 h after transfection. **b** Percentage of Cre-mediated recombination in CD45+ subsets in vivo (*n* = 3, bars represent mean values +/− SD). **c** Percentage of Cre-mediated recombination in CD4+ and CD8+ T cells in vivo (*n* = 3, bars represent mean values +/− SD). Statistical significance was calculated using one-way ANOVA with Tukey's multiple comparisons test. Cell icons were created with BioRender.com. **d** Schematic representation of mRNA delivery in vivo to assess T-cell-specific luciferase expression. Created with BioRender.com. **e** Quantitation of luminescence signal in untreated (Unt.) splenic T cells or 6 h after transfection of 5 μg mRNA luciferase with ONA or bAC-7c (*n* = 3, bars represent mean values +/− SD). Statistical significance was calculated using one-way ANOVA with Dunnett's multiple comparisons test. **f** Effect of mRNA dose in isolated splenic T cells 6 h after transfection with bAC-7c (*n* = 3, bars represent mean values +/− SD).

similar 24 h before and after bAC CART delivery (Supplementary Fig 14c). These results demonstrate that bAC CARTs are promising vectors for in vivo delivery. They are well-tolerated, exhibit spleen tropism without the need for targeting ligands, and can mediate efficient mRNA delivery to T lymphocytes.

## Discussion

We designed, synthesized, and characterized a library of 24 bAC CARTs and identified several that exhibit efficient primary T-cell transfection in vitro and selective spleen tropism in vivo, outperforming ONA and MTC CARTs. A further advantage of these bAC CART delivery systems is that they are readily assembled through a step economical organocatalytic ring-opening oligomerization and deprotection, a two-step process that allows for rapid exploration of variations in initiators, lipid type and block length, charge block length and CART-cargo ratios. bAC CART/mRNA complexes are readily formed by mixing the carrier and cargo components and exhibit a size range (130–280 nm) suitable for various applications. bAC CART complexes are stable upon formulation at pH ≤ 5.5 but readily release their polyanionic cargo at physiological pH.

T cells represent important mediators of immune protection, and are a key target for cell reprogramming, vaccination, and immunotherapy. Notwithstanding their clinical importance, efficient and selective delivery systems for T-cell transfection are still lacking, prompting our investigation of whether the CARTs could address this need. Like some LNP delivery systems, polycationic CARTs form

electrostatic complexes with polyanionic cargos, protecting the latter while allowing for cell entry. A unique feature of the CART system is its ability to irreversibly convert to neutral byproducts through an O-to-N acyl shift, thereby releasing its polyanionic cargo. CARTs have been shown to efficiently transfect different cell types in vivo, leading to preclinical candidates for COVID and cancer vaccines[13]. Compared to previous CART systems, bAC CARTs possess a distinct polymeric backbone and side chain spacing and exhibit highly effective T-cell delivery in vitro, outperforming current transfection methods such as electroporation and lipid-mediated delivery reagents in our comparative studies. We also demonstrated that bAC CART delivery mediates efficient ex vivo CAR-T cell engineering, and the resulting engineered CAR-T cells exhibit antigen-specific killing. While all CARTs systems are candidates for preclinical advancement, ONA CART employs lipids that contain unsaturated double bonds (oleyl, nonenyl), which could be susceptible to oxidation or isomerization during extended storage or interfere with acid-catalyzed removal of the Boc protecting group as required for the ONA CART preparation. In contrast, neither of the leading bAC candidates (bAC-4b, bAC-7c) contain unsaturated lipids, making them easier to access synthetically and to formulate, and potentially more stable for storage and clinical use.

Novel LNP structures and formulations for non-liver delivery have been recently discovered[33,47,48]. The emergence of LNP delivery systems that exhibit organ tropism without a targeting ligand opens significant new opportunities for RNA and DNA delivery. Notably, we showed that bAC CARTs exhibit nearly exclusive spleen tropism (97%)

without the use of targeting ligands and enable efficient transfection of B and T lymphocytes in vivo, which is distinct from many LNP delivery systems targeting liver[13,49,50]. Since the spleen is the major secondary lymphoid organ, the exclusive spleen tropism of bAC CARTs may be highly favorable for biomedical applications such as T-cell immunotherapy while minimizing off-target effects.

In conclusion, we demonstrate that changes in the composition of the CART polymeric backbone and spacing of side chains result in improved transfection of primary T lymphocytes in vitro and highly selective organ tropism in vivo without the need for targeting ligands. Further directions include the use of bAC CARTs to deliver other polyanions such as siRNA, saRNA and circRNA for both research and therapeutic applications.

## Methods
### CART co-oligomer synthesis
The synthesis of all monomers are described in Supplementary Information. All CARTs were prepared without a glove box using standard Schlenk line techniques. A representative procedure to synthesize Boc-protected CART-1a starts by adding bAC-A monomer (49.27 mg, 17 equiv, 157 μmol) and thiourea catalyst (7.00 mg, 19.76 μmol, TU, TU = 1-(3, 5-bis(trifluoromethyl)phenyl)−3-cyclohexylthiourea)) into a glass GPC vial with a stir bar. The vial was flushed with nitrogen, and solids were then dissolved in 75 μl DCM. 30 mg/mL benzyl alcohol solution was prepared in a separate vial, and 34.3 μl was added to the monomer solution (1.00 mg, 1 equiv). 3.01 mg of DBU (2 equiv, 19.76 μmol, DBU = 1,8-diazabicyclo[5.4.0]undec-7-ene) was added to start the polymerization. Three hours later, 20.47 mg Boc-morpholinone (11 equiv, 101.72 μmol) was added as a solid. After 3 h, the reaction was quenched with 6 drops of AcOH and transferred to a 2 kDa dialysis bag and dialyzed against MeOH for 4 h. The dialyzed solution was concentrated to yield a clear oil as the product. The block length and number average molecular weight was determined by 1H-NMR end-group analysis. Dispersity (Đ) was determined by gel permeation chromatography (GPC).

Boc groups were removed to yield the deprotected oligomer as the active vehicle as follows. In a 1-dram vial, oligomers (0.5 μmol) were deprotected in a trifluoroacetic acid/ distilled DCM solution (1:4 v/v, 500 μL) under slow stirring and ambient atmosphere for 2 h. Solvent was removed in vacuo and the samples were stored under high vacuum for 18 h. The deprotected oligomer, as a thin film, was dissolved in DMSO to achieve 30 mM cation concentrations and stored at −20 °C ready to use.

### Dynamic light scattering (DLS) for size analysis and Zeta potential for surface charge analysis
CARTs were added to PBS 5.5 containing 420 ng of eGFP mRNA (Trilink L-7601) at 10:1 charge ratio (cation: anion). The formulations were mixed for 20 s using a micropipette (drawing and dispensing 100 μl twice/second). The formulations were then immediately transferred to a disposable clear plastic cuvette, and the sizes measured by Nano-Brook Omni (Serial No: 280097). Surface charge was monitored at 0, 10, 20, 30 min after formulation using the same instruments.

### mRNA delivery in Jurkat cell studies
Jurkat cell line Clone E6-1 was acquired from ATCC (TIB-152). Cells were maintained in suspension culture with RPMI 1640 medium supplemented with 2 mM glutamine, 10% FBS, and 1% penicillin−streptomycin. Cells were sub-cultured every two days. All cell lines were kept in culture at 37 °C in a humidified incubator with 5% CO₂, and regularly tested for mycoplasma contamination (Lonza[46,47]). Before CART transfection Jurkat cells were washed twice with serum-free media and resuspended in serum-free medium at $4 \times 10^6$ cells/ml, 25 μl of cell suspension were added into a 96-well round bottom plate (100,000 Jurkats per well). The following CART transfection protocol of eGFP mRNA was performed

using the described amount (100 or 200 ng) of mRNA per condition. CARTs were complexed with 420 ng of mRNA in PBS (pH 5.5) to a total volume of 8.4 μl at a 10:1 N:P ratio. The formulations were mixed for 20 s using a micropipette and immediately 2 μl was transferred to each well (100 ng per well, $n = 4$). After 2−4 h, 150 ul of complete media was added to each well. Flow cytometry was performed 24 h after transfection. Viability was measured with Near-IR viability dye (Invitrogen L34992). Data were collected on an Attune Nxt Flow Cytometer.

Jurkat cells were also used to test the stability of CARTs complexed with mRNA. CARTs were complexed with mRNA in PBS (pH 5.5) with 20% sucrose to a total volume of 16.8 μl at a 10:1 N:P ratio. The formulations were immediately frozen in dry ice and transferred to −80 °C for long-term storage. After seven days, formulations were thawed, and 8 μl were transferred to each well (96-well round bottom plate, 200 ng/100,000 Jurkats per well, $n = 4$). In parallel, the same formulations were newly prepared, and 8 μl were immediately transferred to each well ($n = 4$) to compare the transfection efficiency between fresh formulation (prepared immediately before transfection) and the frozen formulation (prepared in advance and stored at −80 °C for one week). After 2−4 h, 150 μl of complete media was added to each well. Flow cytometry was performed 24 h after transfection. Viability was measured with Near-IR viability dye (Invitrogen L34992). Data were collected on an Attune Nxt Flow Cytometer.

### mRNA delivery in human primary T cells
Fresh PBMCs were obtained from de-identified healthy donors at the Stanford Blood Center. Human PBMCs were isolated from whole blood by density gradient centrifugation using SepMate tubes (StemCell Technologies) with Lymphoprep following manufacturer instructions. PBMC's were counted and resuspended in CryoStor CS10 freezing medium (StemCell Technologies) at $1 \times 10^7$ cells/ml for long-term storage in liquid nitrogen. PBMCs were thawed and counted on the day of transfection to isolate total human T cells by following manufacturer instructions from Pan T cell isolation kit, human (Miltenyi Biotec). Isolated T cells were counted and activated with Dynabeads Human T-Activator CD3/CD28 (Gibco). Briefly, in a 24-well plate, isolated T cells were activated with 25 ul of Dynabeads for every $1 \times 10^6$ T cells in 1 ml of complete medium (RPMI 1640 medium containing 10% FBS, 1% penicillin/streptomycin, 50 μM β-mercaptoethanol, and 1% L-glutamine). Dynabeads were removed after the indicated timepoints (6 h, 24 h, and 48 h) using DynaMag magnet, and T cells were washed twice with serum-free RPMI medium. After counting, cells were resuspended in serum-free medium at $4 \times 10^6$ cells/ml, and 25 μl of cell suspension were added into a 96-well round bottom plate (100,000 T cells per well). The CART transfection protocol of eGFP mRNA was then followed as described in the Jurkat transfection section, using the indicated amount of mRNA per condition (50 ng, 100 ng, 200 ng, and 400 ng). CARTs were complexed with mRNA in PBS (pH 5.5) at a 10:1 N:P ratio (formulations with 4:1 and 25:1 charge ratios were also tested). After 2−4 h, 150 μl of complete media was added to each well. Flow cytometry was performed 24 h after transfection, distinguishing CD4+ (BV650 anti-human CD4, BD 563876, 1:300 dilution) and CD8 + T cells (PerCP-Cy5.5 anti-human CD8, BD 560662, 1:200 dilution) by surface marker staining. Viability was measured with Near-IR viability dye (Invitrogen L34992). Data were collected on an Attune Nxt Flow Cytometer.

Primary T cells were also transfected with several commercial reagents. TransIT-mRNA transfection kit (Mirus Bio MIR 2225), Lipofectamine 2000 Transfection Reagent (Invitrogen 11668030), Lipofectamine 3000 Transfection Reagent (Invitrogen L3000008) or Invitrogen™ Neon™ Transfection System 10 μL Kit (Invitrogen MPK1096). Electroporation of RNA was performed following the manufacturer's instructions for primary T cells (program #24: 1600 V/10 ms/3 pulses). All experiments were performed in parallel using 100 ng of mRNA and 100,000 cells per condition from the same

batch of activated primary T cells (24 h activation). Flow cytometry was performed 24 h after transfection, distinguishing CD4+ (BV650 anti-human CD4, BD 563876, 1:300 dilution) and CD8 + T cells (PerCP-Cy5.5 anti-human CD8, BD 560662, 1:200 dilution) by surface marker staining. Viability was measured with Near-IR viability dye (Invitrogen L34992). Data were collected on an Attune Nxt Flow Cytometer.

## CD8 T-cell isolation and stimulation
Peripheral blood mononuclear cells (PBMCs) of healthy donors were isolated from leukoreduction systems (LRS) chambers that were obtained from the Stanford Blood Center. PBMCs were cryopreserved with FBS containing 10% DMSO in liquid nitrogen. CD8 T cells were isolated from the thawed PBMCs with an EasySep human CD8 + T cell isolation kit (Stemcell Technologies) and stimulated with 25 μl Dynabeads Human T-Activator CD3/CD28 (Gibco, 11131D) per $1 \times 10^6$ CD8 T cells in 1 ml of complete RPMI 1640 medium [RPMI 1640 medium (Gibco, 21870092) supplemented with 1× PSA (Gibco, 15240062), 2 mM L-glutamine (Gibco, 25030081), and 10% FBS (Corning, 35-016-CV)] in each well of a 24-well plate for 18 h. CD8 T cells were collected and Dynabeads were removed with a magnet before CART transfection.

## CART transfection of anti-hCD19 mRNA
Activated CD8 T cells were washed with serum-free RPMI 1640 medium supplemented with 1X PSA and 2 mM L-glutamine twice, resuspended with serum-free RPMI 1640 medium, and plated in a U-bottom 96-well plate at a density of $1 \times 10^5$ cells per 25 μl per well. Formulation of CART/mRNA complex was performed by mixing bAC-7c and ONA with mRNA that encode anti-human CD19-41BB-CD3ζ CAR (hCAR) (a generous gift from Ronald Levy Lab in Stanford University)[34,51] or mCherry (TriLink L-7203) in PBS (pH 5.5). CART and mRNA were complexed at charge ratio 10:1 except that bAC-7c and anti-hCD19 CAR mRNA were complexed at an optimized charge ratio 4:1. Formulated CART/mRNA complex was added into each well of CD8 T cells rapidly at a dosage of 200 ng mRNA per well. Same volume of PBS (pH 5.5) was added into untransfected cells as a control. Medium was changed to serum-containing medium by adding 175 μl of RPMI 1640 medium supplemented with 11.4% FBS, 1× PSA, and 2 mM L-glutamine 2 h after transfection and the cells were cultured for another 18 h. Transfection efficiency was evaluated by staining of anti-hCD19 CAR with soluble human CD19-FITC (Acrobiosystems, CD9-HF2H2, 1:160 dilution) using flow cytometry.

## Co-culture of CD8 T cells and Nalm6 cells
Transfected CD8 T cells were collected from the 96-well plate and counted. Wild-type (WT) and CD19 knock-out (KO) Nalm6-GL cells that constitutively express GFP and firefly luciferase (generous gifts from Crystal Mackall Lab in Stanford University)[52,53] cultured in complete RPMI 1640 medium were collected from T75 flasks and counted. CD8 T cells and Nalm6-GL cells were co-cultured in a U-bottom 96-well plate at the indicated effector-to-target (E:T) ratios starting with $6.7 \times 10^4$ cells in total in 200 μl complete RPMI 1640 medium per well for 13 h. Cells were stained with eFluor 780 fixable viability dye (Invitrogen, 65-0865-14), as well as anti-human CD3-Brillian Violet (BV) 421 (clone OKT3, BioLegend, 1:250 dilution) and anti-human CD8-APC (clone SK1, BioLegend, 1:250 dilution), for flow cytometry analysis. Samples were analyzed with Aurora flow cytometer (Cytek Biosciences) and cell populations were characterized with FlowJo 10.9.0. To evaluate the killing of Nalm6-GL cells, number of viable Nalm6-GL cells (e780lowCD3-CD8-GFP + ) in each sample was quantified by adding precision count beads (BioLegend, 424902) into the flow cytometry samples before sample analysis and calculated as Nalm6-GL cell count x precision count beads volume (μl) x precision count bead concentration as in stock (beads/μl)/precision count beads count.

Killing of Nalm6-GL cells was calculated as the reduction of the number of Nalm6-GL cells in each co-culture group compared to Nalm6-GL cells alone.

For the evaluation of functional markers in CD8 T cells, anti-human CD107a-BV711(clone H4A3, BioLegend, 1:800 dilution), brefeldin A (eBioscience 00-4506-51) and monensin (eBioscience 00-4505-51) were added to the cells along with the co-culture. Cells were stained with anti-human CD3 and anti-human CD8 antibodies as described above, fixed with BD FACS lysing solution (BD Biosciences, 349202), permeabilized with BD FACS permeabilizing solution II (BD Biosciences, 340937), and stained for IFN-γ and TNF-α with anti-human IFNγ-BV785 (clone 4 S.B3, BioLegend, 1:200 dilution) and anti-human TNFα-BV650 (clone MAb11, BioLegend, 1:800 dilution). Expression of anti-hCD19 CAR was detected with soluble human CD19-FITC (Acrobiosystems, CD9-HF2H2, 1:160 dilution). The expression of functional markers (CD107a, IFN-γ and TNF-α) were analyzed in the cells with or without anti-hCD19 CAR expression in the CD8 T cells (CD3 + CD8 + ) transfected with anti-hCD19 CAR mRNA as well as in mCherry+ and mCherry- CD8 T cells in the mCherry mRNA transfected group and in total cells of untransfected group.

## Cy5-labeled mRNA uptake and translation by flow cytometry
Cy5-labeled mRNA was generated with the Label IT Nucleic Acid Labeling kit, Cy5 (Mirus Bio MIR3700) following manufacturer instructions. A 1:1 mix of Cy5-labeled and unlabeled mRNA encoding eGFP was complexed with CARTs, using 200 ng of mRNA per condition, and delivered to either Jurkat or primary T cells. Both Cy5-labeled mRNA uptake and eGFP translation were measured at different timepoints (30 min, 2 h, and 24 h) by flow cytometry. Viability was measured with Near-IR viability dye (Invitrogen L34992). Data were collected on an Attune Nxt Flow Cytometer.

## Confocal microscopy
Cy3-labeled mRNA was generated with the Label IT Nucleic Acid Labeling kit, Cy3 (Mirus Bio MIR3600) following manufacturer instructions. 200 ng of Cy3-labeled mRNA was complexed with CARTs and delivered to Jurkat cells. After 2 or 6 h, cells were washed twice with PBS and fixed with 4% paraformaldehyde in PBS for 20 min at room temperature to preserve cellular structures and mRNA localization. Subsequently, cells were rinsed three times with PBS to eliminate any traces of fixative. Hoechst 33342 staining was then used to label cell nuclei, and the cells were washed again with PBS to remove excess staining dye. Cytospin was used to create a thin and even monolayer of fixed cells for microscopy analysis. Approximately 100 μl of cell suspension was aliquoted into each Cytospin well and centrifuged at maximum speed for 3 min. Cells were imaged on a Zeiss LSM 880 confocal laser scanning microscope using a ×63 oil objective lens. The percent area of each cell expressing Cy3 was calculated in ImageJ using nuclei staining to approximate Jurkat cell boundaries. Binary thresholds were applied to the red and blue channels to identify Cy3-labeled mRNA and Hoechst-labeled nuclei, respectively, and the borders of each nucleus delineated with the wand tool. Within each nucleus, the Cy3+ area was then measured, with more or less Cy3+ area attributed to more diffuse versus more localized expression. To ensure representative results, we analyzed at least 20 cells from 5 different fields for each experimental condition.

## RNA release assay by Qubit RNA HS dye
The 1× Qubit RNA HS solution was prepared in water from a 200× stock solution provided by the vendor (Thermo Fisher Q32852). CART/ luciferase mRNA complexes were formulated as described above. To measure Qubit fluorescence at 0 min, 4.2 μl of formulation was immediately diluted in 75 μl water and pipetted to mix well. In all, 20 μl of the dilution was resuspended in 180 μl 1× Qubit solution and fluorescence measured by a Qubit fluorometer. To measure other

timepoints, the same procedure was followed except for replacing water with PBS 7.4 to allow CARTs to charge-alter. Fluorescence was measured at 1 min, 2.5 min, 5 min, 10 min, and 15 min after PBS 7.4 addition, with mRNA release calculated as:

$$\text{mRNA release}(\%) = \frac{\text{Fluorescence}(\text{Time}_X) - \text{Fluorescence}(\text{no mRNA})}{\text{Fluorescence}(\text{free mRNA}) - \text{Fluorescence}(\text{no mRNA})}$$

### Luciferase mRNA biodistribution

C57BL/6 J (#000664), BALB/cJ (#000651), and Ai14 (#007908) were purchased from Jackson Laboratory. All mice were 6 to 8 weeks old at the time of the experiments. All mice in this study were maintained under specific pathogen-free conditions, a 12-h light/12-h dark cycle, and temperatures of -18–23 °C with 40–60% humidity. All mice were handled according to the approved institutional animal care and use committee (IACUC) protocols of Stanford University.

Female BALB/c mice were purchased from Jackson Laboratory and housed in the Laboratory Animal Facility of the Stanford University Medical Center. CARTs were complexed with 5 μg of luciferase mRNA (Trilink L-7202) in PBS 5.5 to a total volume of 100 μl at 10:1 charge ratio (formulations with other charge ratios were also tested), with the same formulation procedure described above. The formulations were mixed for 20 s using a micropipette (drawing and dispensing 100 μl twice/second) before retro-orbital injections. Six hours after injection, D-luciferin solution (100 μl, 30 mg/mL) was injected intraperitoneally, and luminescence was measured using AMI Imaging system charged-coupled device camera and analyzed with Aura. Spleen, liver, and lung were also imaged as isolated organs.

Isolated spleens were additionally processed to obtain single-cell suspensions. Spleen tissues were gently mashed to achieve a homogenous cell population. To remove red blood cells, suspensions were subjected to red blood cell lysis. Total T cells were isolated from the cell suspension using the Pan T Cell Isolation Kit II, mouse (130-095-130) following manufacturer instructions. Following cell counting, 2 million T cells per well were plated in a 96-well format. Subsequently, D-luciferin was added to each well, and the plate was immediately read for luminescence using a BioTek Microplate Reader.

### Cell specificity of mRNA delivery with CART

To determine the cell specificity of mRNA delivery with CART, mice were injected with 7.5 μg of Cy5-labeled mRNA encoding luciferase. After 2 h, spleens were isolated and processed into a single-cell suspension, followed by red blood cell lysis. Single-cell suspensions were stained with Zombie NIR (BUV570, BioLegend 423103) to exclude dead cells and labeled with antibodies against various cell surface markers: anti-CD45 (clone 145-2C11, BioLegend, 1:300 dilution) for total leukocytes, anti-CD8α (clone 53-6.7, BioLegend, 1:200 dilution) for CD8 + T cells, anti-CD4 (clone RM4-5, BioLegend, 1:200 dilution) for CD4 + T cells, anti-CD11c (clone IM7, BioLegend, 1:400 dilution) for dendritic cells, anti-CD19 (clone 30-F11, BioLegend, 1:200 dilution) for B cells, and anti-F4/80 (clone H1.2F3, BioLegend) for macrophages. After staining, cells were washed twice and subjected to flow cytometry analysis. Data acquisition was performed using an Attune Nxt Flow Cytometer. The percentage of Cy5+ cells per cell type was then calculated.

### Ai14 Cre-mediated recombination

Female Ai14 mice were purchased from Jackson Laboratory and housed in the Laboratory Animal Facility of the Stanford University Medical Center. CARTs were complexed with 15 μg Cre mRNA (Trilink L-7211) in PBS 5.5 to a total volume of 100 μl at 10:1 N:P ratio, with the same formulation procedure described above. Spleens were isolated 48 h after transfection to measure Cre-mediated recombination.

Spleens were smashed with a 100-μm strainer to make a single-cell suspension. Red blood cells were lysed before staining. Single-cell samples were then stained with Zombie NIR (BUV570, BioLegend 423103), anti-CD45 (clone 145-2C11, BioLegend, 1:300 dilution), anti-CD8α (clone 53-6.7, BioLegend, 1:200 dilution), anti-CD4 (clone RM4-5, BioLegend, 1:200 dilution), anti-CD11c (clone IM7, BioLegend, 1:400 dilution), anti-CD19 (clone 30-F11, BioLegend, 1:200 dilution), and anti-F4/80 (clone H1.2F3, BioLegend, 1:100 dilution). Cells were then washed twice and analyzed by flow cytometry. Data were collected on an Attune Nxt Flow Cytometer.

### Preclinical toxicology analysis

Mice were dosed with 7.5 μg of luciferase mRNA complexed with bAC CART or LNP. Blood samples were collected from the mice 24 h after treatment. Around 500 μl of blood per sample were submitted to the Stanford Diagnostics Lab for hematological analysis. Assessed parameters included Complete Blood Count (CBC) with white blood cell differential, electrolytes, liver function tests, kidney function tests, and glucose. Blood samples collected 24 h after CART/mRNA injection were also processed to isolate serum for Luminex analysis conducted by the Human Immune Monitoring Center at Stanford University. This analysis was performed using the Mouse 48 plex Procarta kit (Thermo Fisher/Life Technologies) following manufacturer instructions.

### Statistical analysis

All statistical analyses were performed with Prism (GraphPad Software 9.2.0). For comparing more than two groups, two-way ANOVAs were applied. Differences between groups were considered significant for $P$ values < 0.05. No statistical methods were used to predetermine sample sizes. Mice were assigned to the various experimental groups randomly. Data collection and analysis were not performed blind to the conditions of the experiments.

### Reporting summary

Further information on research design is available in the Nature Portfolio Reporting Summary linked to this article.

## Data availability

The experiment data that support the findings of this study are available from the corresponding author upon request. All the data generated in this study are provided in the Source Data file, including data for the supplementary information. Source data are provided with this paper.

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

## Acknowledgements

Supported by NIH R35-CA209919 (H.Y.C.), NIH 5R01CA245533-03 (P.A.W. and R.M.W.), NSF-CHE 2002933 (R.M.W.), Emerson Collective (H.Y.C. and P.A.W.), the Stanford Center for Molecular Analysis and Design (Z.L.), and Stanford Bio-X (L.A.). We thank the technical support from the Veterinary Service Center (VSC) and the Human Immune Monitoring Center (HIMC) at Stanford Medicine. H.Y.C. is an Investigator of the Howard Hughes Medical Institute. We thank Harrison P. Rahn for providing hydrogenated farnesol lipids for bAC-4 synthesis, Ronald Levy Lab for providing anti-hCD19 mRNA, Crystal Mackall Lab for providing Nalm6-GL cell lines, and Uma Mangalanathan for assisting with the anti-hCD19 CAR and Nalm6-GL co-culture experiment. C.A.B. is supported by the Chan Zuckerberg Biohub, grant 5DP1DA04608902 from the National Institute on Drug Abuse, NIH, grant 1016687 from the Burroughs Wellcome Fund Investigators in the Pathogenesis of Infectious Diseases, and a Bill & Melinda Gates Foundation Pilot Grant through the Stanford Human Systems Immunology Center. C.A.B. is the Tashia and John Morgridge Faculty Scholar in Pediatric Translational Medicine from the Stanford Maternal Child Health Research Institute.

## Author contributions

Z.L., L.A. and P.A.W. conceived the experiments. Z.L., L.A., R.P., S.K.W. and A.R. performed the experiments. R.M.W., C.A.B. and H.Y.C. provided guidance on the work. Z.L., L.A. and P.A.W. wrote the manuscript with input from all co-authors.

## Competing interests

H.Y.C. is a co-founder of Accent Therapeutics, Boundless Bio, Cartography Biosciences, Orbital Therapeutics, and an advisor of 10x Genomics, Arsenal Biosciences, Chroma Medicine, and Spring Discovery. P.A.W. is a co-founder of BryoLogyx and N1 Life and an advisor to BryoLogyx, N1 Life, Synaptogenix, Cytokinetics, Evonik, SuperTrans Medical, Ativo, and Vault Pharma. R.M.W. is an advisor to Evonik. The remaining authors declare no competing interests. C.A.B. is a scientific advisory board member for Catamaran Bio, Immunebridge, and Qihan Ltd on topics unrelated to this study.
