## [Peer Review File · Nature Communications]

Charge-altering releasable transporters enhance mRNA delivery in vitro and exhibit in vivo tropismREVIEWER COMMENTS

Reviewer #1 (Remarks to the Author):

The authors have previously published on a delivery system for mRNA named charge-altering releasable transporters (CARTs) and in this manuscript disclose a further iteration of this delivery class. While a previously identified CARTs incorporating both oleyl and nonenyl lipids termed Oleyl-Nonenyl Amino CARTs, i.e., ONA showed good (up to ~80%) transfection efficiency in Jurkat cells, these CARTs were less effective (10-20%) in the transfection of primary T cells (PMID: 29891683). Now the authors report a new CART delivery system with a beta-amido carbonate (bAC) backbone that showed improved in vitro transfection efficiency in primary T cells (up to 70%). After systemic delivery in mice the best performing bAC CARTs yielded up to 97% spleen tropism (i.e. bAC-4b) and transfected 8% primary splenic T cells (i.e. bAC-7c).

This new delivery class could have potential for in situ T cell engineering, e.g. for in situ chimeric antigen receptor T cell generation. However, some issues need to be addressed before this manuscript can be considered for publication.

- 1) The enhanced transfection in primary T cells with the bAC CARTs is relevant for ex vivo T cell manipulations. However, the authors could have further investigated the effects of the CARTs on T cell activation and functionality.
- 2) Can the authors further explain how the faster change of surface charge with the bAC CARTs results in an earlier onset of protein expression, as there was no faster mRNA release? Please extend on this in the discussion section.
- 3) The bAC-4b possess a higher in vivo transfection efficiency in spleen than ONA CARTs evidenced by the use of luciferase mRNA as reporter gene, while bAC-7c showed equal levels of transfection than ONA CARTs. In the Ai14 reporter mouse model, bAC-7c achieved two fold higher percentages of Cre-mediated recombination of primary T cells (~8%) than bAC-4b (~4%). It would be interesting to see how the bAC CARTs relate to the ONA format in transfecting splenic T cells (using Cre recombinase mRNA or by testing another reporter-mRNA, e.g. eGFP or mCherry). As the authors also highlight the A14 model only provides information on the percentage of cells but not on protein expression levels. Taken together, the presented in vivo data is not enough to appreciate or demonstrate a better potential of the bAC CARTs over the previously identified ONA CARTs.
- 4) Line 309: "They are well-tolerated". How was this tested? Can the authors add data to make this claim?
- 5) The discussion could be potentially improved by providing some additional information on the therapeutic potential of the bAC CARTs, e.g. ex vivo T cell engineering and in situ chimeric antigen receptor-T cell generation.

Reviewer #2 (Remarks to the Author):

The authors reported on the polymer-based material for the mRNA delivery to the T cells. The in vitro transfection activity to the primary T cell and in vivo T cells in spleen is demonstrated.

While the manuscript includes improvement in terms of the mRNA delivery to the T cells, the proceeding is not sufficient to be accepted in Nature communication.

1. The authors show the previous strategy and materials in Figure 1a to 1c, this information is confusing since they make unclear what is a key material in this manuscript. Figure 1d should be more focused.
2. The most mysterious point is that the mRNA was coated in acidic pH, but once exposed in neutral pH, the mRNA was released quite rapidly within 5 min (Fig 4d). Why the mRNA can be delivered to the spleen when mRNA was released from the particle in physiological pH in blood.
3. In relation to the comment 2, the interaction of their polymer and mRNA in blood should be monitored (i.e. FRET between polymer and mRNA).
4. The intra-spleen distribution of mRNA should be monitored.
5. The Figure 5g showed that the mRNA was expressed in B cells, macrophage, as well as T cells. The expression in B cells and macrophage should be more focused. Also, more quantitative comparison using luciferase or GFP should be demonstrated since Ai14 mouse cannot compare the transfection efficiency quantitatively.
6. The most critical point is that no therapeutic effect data is included in this manuscript. To claim the proceeding of their material, therapeutic data should be shown to be accepted in Nature Communication.

Reviewer #3 (Remarks to the Author):

Li et al developed novel CART formulations for T cell targeting. The topic is important, the manuscript is well-written, the data are solid. The overall concept is not brand new given that the authors have previously published studies on CARTs, this study developed more efficient CART delivery of mRNA to spleen/T cells.

Comments/requests:

1. In the introduction/discussion the authors should mention/better discuss organ/cell type targeting LNPs developed by others (Dan Siegwart, Katie Whitehead etc..).
2. Experiments with Cre mRNA (Fig. 5): it would be great if the authors could add a dose response study and look at spleen/T cell targeting with 5, 15 and 45 ug Cre mRNA. Then, using the optimal dose, it

would be nice to see T cell targeting after 1, 2 or 3 injections of Cre mRNA.

3. Line 309: the authors claim that the CART platform is well-tolerated in mice. How was this measured in the study? To support this claim it would be great to look at potential inflammatory responses induced by CARTs (try it with unloaded and mRNA-loaded particles). IL-6 and type I IFN measurements (4 and 24 hs post CART administration) in the serum would provide some information. Humans tolerate inflammation significantly less than mice (PMID: 35332327) so it would be important to check the ability of CARTs to activate the immune system. Upon strong immune activation, application of CARTs in humans is questionable due to potential toxicity issues.

Response to reviewers' comments

We thank the reviewers for their time and effort in carefully reviewing our manuscript, and for their insightful and constructive feedback. We have tried to be responsive to all of their requests. The request for information of therapeutic potential resulted in the addition of two new co-authors. Please see our responses to individual comments below.

REVIEWER COMMENTS

Reviewer #1 (Remarks to the Author):

The authors have previously published on a delivery system for mRNA named charge-altering releasable transporters (CARTs) and in this manuscript disclose a further iteration of this delivery class. While a previously identified CARTs incorporating both oleyl and nonenyl lipids termed Oleyl-Nonenyl Amino CARTs, i.e., ONA showed good (up to ~80%) transfection efficiency in Jurkat cells, these CARTs were less effective (10-20%) in the transfection of primary T cells (PMID: 29891683). Now the authors report a new CART delivery system with a beta-amido carbonate (bAC) backbone that showed improved in vitro transfection efficiency in primary T cells (up to 70%). After systemic delivery in mice the best performing bAC CARTs yielded up to 97% spleen tropism (i.e. bAC-4b) and transfected 8% primary splenic T cells (i.e. bAC-7c).

This new delivery class could have potential for in situ T cell engineering, e.g. for in situ chimeric antigen receptor T cell generation. However, some issues need to be addressed before this manuscript can be considered for publication.

1) The enhanced transfection in primary T cells with the bAC CARTs is relevant for ex vivo T cell manipulations. However, the authors could have further investigated the effects of the CARTs on T cell activation and functionality.

We appreciate the reviewer's point about the state of activation and functionality of T cells after CART transfection. We have now measured the levels of activation markers CD25 and CD69, and exhaustion marker PD1. Our results show no difference in phenotype and activation state between CART transfected T cells and untreated control.

Figure S5a:

Furthermore, we also measured the capacity of transfected T cells to produce effector cytokines IFN γ and TNF- α and observed no significant differences between bAC-7c and untreated samples.

Figure S5b:

Finally, we also measured the T cell proliferation potential after non-specific activation with anti-CD3/CD28 and IL-2 cytokine and showed no significant differences in cell growth and viability after CART transfection.

Figure S5c:

These new data showing the maintenance of phenotype and functionality of human T cells after transfection with bAC-7c CART have been incorporated as Supplementary figure 5.

2) Can the authors further explain how the faster change of surface charge with the bAC CARTs results in an earlier onset of protein expression, as there was no faster mRNA release? Please extend on this in the discussion section.

We conducted biophysical characterization of bAC CARTs/mRNA complexes (size, surface charge, mRNA release) which provided useful comparative information on the properties of nanoparticles themselves in a standard buffer. However, these analyses are done in a non-cell context (buffers), while cellular transfection occurs in a different, dynamic and much more complicated environment. Thus, while the biophysical characterization of the starting CART complexes allows for systematic comparisons of size and charge-change in buffers, it would not be expected to correlate and often does not correlate with cellular transfection in vitro due differences in the chemical environment (buffer vs. cell culture, pH differences). Such variations are also widely observed and documented in attempts to correlate in vitro transfection with in vivo transfection due again to differences in microenvironment (e.g. corona effects).

With the above noted but with an effort to gain deeper insights into the potential reasons for the earlier onset of protein expression in bAC CARTs, we employed confocal microscopy to track Cy3-labeled mRNA delivered by CARTs. bAC-7c exhibited notably more localized mRNA distribution compared to ONA CART and MTC-7c in Jurkat cells. This correlation aligned with the observed trend in protein expression onset. These new data were incorporated as Supplementary Figure 9.

Figure S9:

3) The bAC-4b possess a higher in vivo transfection efficiency in spleen than ONA CARTs evidenced by the use of luciferase mRNA as reporter gene, while bAC-7c showed equal levels of transfection than ONA CARTs. In the Ai14 reporter mouse model, bAC-7c achieved two fold higher percentages of Cre-mediated recombination of primary T cells (~8%) than bAC-4b (~4%). It would be interesting to see how the bAC CARTs relate to the ONA format in transfecting splenic T cells (using Cre recombinase mRNA or by testing another reporter-mRNA, e.g. eGFP or mCherry). As the authors also highlight, the A14 model only provides information on the percentage of cells but not on protein expression levels. Taken together, the presented in vivo data is not enough to appreciate or demonstrate a better potential of the bAC CARTs over the previously identified ONA CARTs.

We have now included a direct comparison of bAC CARTs and ONA with our Ai14 model and showed a similar proportion of T cells are transfected in vivo (now Figure 7b). While we had bAC-4b and bAC-7c comparisons, now we include ONA in the updated figures 7b and 7c.

Figure 7b:

Figure 7c:

Furthermore, we have introduced a novel approach for assessing the levels of protein translation in vivo by directly transfecting T cells with bAC or ONA CART and subsequently measuring luminescence in a plate format post T cell isolation. Our findings indicate that bAC CART exhibited a threefold increase in luciferase expression in splenic T cells compared to ONA CART, as illustrated in Figure 7e (newly incorporated data). This data demonstrates that bAC-7c induces more protein expression in splenic T cells than ONA CART.

Luciferase expression in T cells (Figure 7e):

Additionally, we have elaborated on the benefits of bAC CART compared to ONA CART in the discussion section. Specifically, ONA CART employs lipids that contain unsaturated double bonds (oleyl, nonenyl), which could be susceptible to oxidation during extended storage and from a purely chemical synthesis and manufacturing point of view, they limit and could interfere with acid-catalyzed removal of the Boc protecting group, required for the ONA CART preparation. In contrast, neither of the leading bAC candidates (bAC-4b, bAC7c) contain unsaturated lipids, making them easier to access synthetically, and potentially more stable for storage and clinical use.

4) Line 309: "They are well-tolerated". How was this tested? Can the authors add data to make this claim?

We appreciate the reviewer's comment. We have limited our tolerability studies in mice to acute effects to limit the number of mice used with the expectation that in depth tolerability studies will be best done in advanced animal models due to variation in metabolism, immune response and organ function. However, to be responsive to the reviewer, we have now included more data to demonstrate the safety and tolerability of bAC CARTs. Briefly, we performed a pre-clinical toxicology analysis in blood and demonstrated that i.v. delivery of mRNA with bAC CART results in baseline levels of metabolites and blood cells counts. Compared to the standard LNP, bAC-7c resulted in similar levels of blood components, and liver and kidney waste products.

Blood counts (Figure S14a):

Liver and kidney waste (Figure S14b):

All metabolites measured have been described in the methods section and a summary of the toxicology report has been included as Supplementary Figure 14a and 14b.

5) The discussion could be potentially improved by providing some additional information on the therapeutic potential of the bAC CARTs, e.g. ex vivo T cell engineering and in situ chimeric antigen receptor-T cell generation.

The intent of this work is to address an urgent need for new delivery systems which has resulted in the introduction of a new CART family which exhibits improved gene delivery both in vitro and in vivo. Therapeutic applications are extensive and expensive and best conducted in a more focused future study which is planned with multiple animal models. However, to be responsive to the reviewer's request, we conducted a pilot study using bAC CARTs to engineer chimeric antigen receptors T cells (CAR-T). We have now shown that bAC-7c can effectively generate CD19 CAR-T cells with effector functions, as demonstrated by specific cell killing of Nalm6 cells, and overexpression of degranulation markers and inflammatory cytokines. CD19 CAR-T cells are currently in clinical use for the treatment of cancer (e.g. ALL) and other CD targeting CAR T and CAR NK cells

are being advanced preclinically and clinically. We have added the names of colleagues as coauthors who worked with us to address this therapeutic opportunity.

Anti-hCD19 expression and quantification of live cells after co-culture of CAR T cells and cancer cells:

Figure 4c:

Figure 4d:

Figure 4e:

This new data has now been included as Figure 4b-e, and Supplementary Figure 8. The additional data boosts the impact of the work and demonstrates translational potential of bAC CARTs.

Reviewer #2 (Remarks to the Author):

The authors reported on the polymer-based material for the mRNA delivery to the T cells. The in vitro transfection activity to the primary T cell and in vivo T cells in spleen is demonstrated.

While the manuscript include improvement in terms of the mRNA delivery to the T cells, the proceeding is not sufficient to be accepted in Nature communication.

1. The authors show the previous strategy and materials in Figure 1a to 1c, these information is confusing since they make unclear what is a key materials in this manuscript. Figure 1d should be more focused.

We thank the reviewer. To address the reviewer's confusion, we have modified our introductory Figure 1 to make it more concise. We removed unnecessary information (previously panel 1c) and now we show the whole structure of the MTC and bAC backbones, highlighting the key differences (distinct lipid spacing and backbone) and explicitly mentioning the key advantages of bAC CARTs.

Figure 1c:

2. The most mysterious point is that the mRNA was coated in acidic pH, but once exposed in neutral pH, the mRNA was released quite rapidly within 5 min (Fig 4d). Why the mRNA can be delivered to the spleen when mRNA was released from the particle in physiological pH in blood.

We appreciate the reviewer's point and hope now to make clearer the differences between the methods used to determine relative rates of RNA release. It is important to note that the rates of mRNA release can vary considerably in buffers (used for characterization) and in cellular assays and in vivo assays (used for transfection and translation) as the particles undergo change that is influenced by their chemical microenvironment. In buffers at pH 5.5, the complexes slowly degrade with minimal release of RNA. In contrast, at physiological pH, the mRNA release is faster. The rapid (5 min) release that the reviewer referred to was determined in a physiological buffer (pH 7.4).

It is important to note that mRNA/CART transporter complexes are affected by a variety of factors during characterization, formulation and delivery processes. The chemical microenvironment which influences size, stability and release kinetics is different in buffer, in in vitro cell culture and in in vivo biodistribution studies. For example, the ex vivo buffer environment is homogeneous and constant while the in vitro and in vivo environments are heterogeneous and dynamic as particles translocate through varying barriers. It is also well known that mRNA complexes can rapidly change in vivo due to particle coating with various plasma proteins (known as the protein corona) within 30 seconds of intravenous injection. Similarly, the association of CART/mRNA complexes with endogenous factors (e.g., proteins), and the identity of the resultant corona could influence particle characteristics and organ transfection. This is an active area of investigation that exceeds in scope this introductory study as it is best done with more advanced animal models relevant to human clinical studies. The unique performance of the bAC-CARTs has now justified initiation of these more extensive and expensive studies.

3. In relation to the comment 2, the interaction of their polymer and mRNA in blood should be monitored (i.e. FRET between polymer and mRNA).

We agree that learning more about this delivery system in murine models could be done. However, as noted with respect to comment 2, our efforts, time and resources are better placed in studies involving more advanced animal models which are extensive and expensive. As bAC CARTs deliver mRNA almost exclusively to spleens, we have focused our current characterization on splenocyte transfection. Future studies in advanced animals beyond the initial scope of this study will provide further information on this point.

4. The intra-spleen distribution of mRNA should be **monitored**.

In response to the reviewer on the value of measuring the distribution of mRNA in the spleen after delivery with bAC CARTs, we have, using mRNA covalently labeled with Cy5, measured mRNA spleen-distribution. Briefly, 2 hours after i.v delivery of Cy5-mRNA with bAC-7c, spleens were isolated and the levels of Cy5-mRNA in splenocytes was quantified by flow cytometry. We showed that on average 85 % of Cy5-mRNA is internalized by CD45+ splenocytes (Supplementary Fig. 13a), which is equally distributed among dendritic cells, macrophages, and B cells (20 %). Only a small proportion is localized to CD8+ T cells (5 %) and CD4+ T cells (1 %). This data is now included as Supplementary figure 13.

Figure S13a:

Figure S13b:

5. The Figure 5g showed that the mRNA was expressed in B cells, macrophage, as well as T cells. The expression in B cells and macrophage should be more focused. Also, more quantitative comparison using luciferase or GFP should be demonstrated since Ai14 mouse cannot compare the transfection efficiency quantitatively.

Following the reviewer's suggestion, we have now included in Supplementary Figure 12 the levels of transfection in B cells (CD19+) and macrophages (F4/80+).

Figure S12b:

We acknowledge the limitation of the Ai14 model to quantitatively measure transfection in specific cell types. Thus, we used luciferase expression to measure quantitatively the in vivo transfection of sorted T cells when using bAC CARTs compared to ONA and showed a 3-fold increase in T cells transfection.

Figure 7e:

6. The most critical point is that no therapeutic effect data is included in this manuscript. To claim the proceeding of their material, therapeutic data should be shown to be accepted in Nature Communication.

The intent of this work is to address an urgent need for new delivery systems which has resulted in the introduction of a new CART family which exhibits improved gene delivery both in vitro and in vivo. Therapeutic applications are extensive and expensive and best conducted in a more focused future study which is planned with multiple animal models. However, to be responsive to the reviewer's request, we conducted a pilot study using bAC CARTs to engineer chimeric antigen receptors T cells (CAR-T). We have now shown that bAC-7c can effectively generate CD19 CAR-T cells with effector functions, as demonstrated by overexpression of degranulation markers and inflammatory cytokines, and subsequent antigen-specific killing of Nalm6. CD19 CAR-T cells are currently in clinical use for the treatment of cancer (e.g. ALL) and other CD targeting CAR T and CAR NK cells are being advanced preclinically and clinically. We have added the names of colleagues as coauthors who worked with us to address this therapeutic opportunity.

Anti-hCD19 expression and quantification of live cells after co-culture of CAR T cells and cancer cells:

Figure 4c:

Figure 4d:

Figure 4e:

This new data has now been included as Figure 4b-e, and the gating strategy in Supplementary Figure 8. The additional data boosts the impact of the work and demonstrates translational potential of bAC CARTs.

Reviewer #3 (Remarks to the Author):

Li et al developed novel CART formulations for T cell targeting. The topic is important, the manuscript is well-written, the data are solid. The overall concept is not brand new given that the authors have previously published studies on CARTs, this study developed more efficient CART delivery of mRNA to spleen/T cells.

Comments/requests:

1. In the introduction/discussion the authors should mention/better discuss organ/cell type targeting LNPs developed by others (Dan Siegwart, Katie Whitehead etc..).

To address the reviewer's comments, we have complemented our introduction section by citing recent publications of tissue-specific mRNA delivery. We have also incorporated the following paragraph in the introduction:

Despite the enormous progress, naked mRNA is large and polyanionic, unable to efficiently cross non-polar biological barriers such as the plasma membrane and reach the cytosol to elicit its function in vivo. To realize the full potential of mRNA therapeutics therefore requires efficient delivery technologies. In recent years, significant progress has been made in optimizing lipid nanoparticles (LNPs) for RNA delivery, as evident from the FDA approval of COVID-19 mRNA vaccines [1,2] and multiple therapeutic assets in active clinical trials [3,4,5]. However, LNPs used in clinical trials are mostly restricted to liver delivery when administered intravenously. While there are recent reports on LNP formulations for RNA delivery to the lungs [6,7], lymphatic systems [6,7], and bone marrow [8], no extrahepatic mRNA therapeutics (intravenous) has been thus far for clinical use, highlighting the need for novel delivery systems for organ- and cell-specific mRNA delivery.

And we have now added the corresponding references:

1. Baden, L. R. *et al.* Efficacy and Safety of the mRNA-1273 SARS-CoV-2 Vaccine. *N Engl J Med* **384**, 403–416 (2021).
2. Skowronski, D. M. & De Serres, G. Safety and Efficacy of the BNT162b2 mRNA Covid-19 Vaccine. *N Engl J Med* **384**, 1576–1577 (2021).
3. Gillmore, J. D. *et al.* CRISPR-Cas9 In Vivo Gene Editing for Transthyretin Amyloidosis. *N Engl J Med* **385**, 493–502 (2021).
4. Sahin, U. *et al.* An RNA vaccine drives immunity in checkpoint-inhibitor-treated melanoma. *Nature* **585**, 107–112 (2020).
5. National Library of Medicine (U.S.). (2005, November -). A Study to Evaluate the Safety and Efficacy of mRNA-1345 Vaccine Targeting Respiratory Syncytial Virus (RSV) in Adults ≥ 60 Years of Age. Identifier NCT00103181. <https://clinicaltrials.gov/ct2/show/NCT05127434>
6. Cheng, Q. *et al.* Selective organ targeting (SORT) nanoparticles for tissue-specific mRNA delivery and CRISPR-Cas gene editing. *Nat Nanotechnol* **15**, 313–320 (2020).
7. LoPresti, S. T., Arral, M. L., Chaudhary, N. & Whitehead, K. A. The replacement of helper lipids with charged alternatives in lipid nanoparticles facilitates targeted mRNA delivery to the spleen and lungs. *J Control Release* **345**, 819–831 (2022).
8. Xue, L. *et al.* Rational Design of Bisphosphonate Lipid-like Materials for mRNA Delivery to the Bone Microenvironment. *Journal of the American Chemical Society* **144**, 9926–9937 (2022).

2. Experiments with Cre mRNA (Fig. 5): it would be great if the authors could add a dose response study and look at spleen/T cell targeting with **5, 15 and 45** ug Cre mRNA. Then, using the optimal dose, it would be nice to see T cell targeting after 1, 2 or 3 injections of Cre mRNA.

We thank the reviewer for suggesting these experiments to complement our in vivo studies. Because Ai14/Cre model might not be best for quantifying protein expression (it only provides % of transfection), we opted not to proceed with dosing experiments on Ai14/Cre models. However, to be responsive to the reviewer's request, we performed a dosing experiment (2, 5, 10 ug) by isolating splenic T cells after luciferase mRNA injection and measuring their luciferase activity. We observed dose-dependent luciferase activity in this experiment, indicating splenic T cell transfection could be boosted with higher mRNA dose. We chose to limit

our dose to 10 ug since it already exceeds the clinically relevant dose of 0.3mg/kg (6 ug mRNA for 20 g mice). The data is now included as Figure 7f.

Figure 7f:

Additionally, we have conducted up to five successive administrations of luciferase mRNA at intervals of 5-7 days and have not detected any alterations in bioluminescence levels. This suggests that protein expression is transient and there are no acute tolerability concerns.

3. Line 309: the authors claim that the CART platform is well-tolerated in mice. How was this measured in the study? To support this claim it would be great to look at potential inflammatory responses induced by CARTs (try it with unloaded and mRNA-loaded particles). IL-6 and type I IFN measurements (4 and 24 hs post CART administration) in the serum would provide some information. Humans tolerate inflammation significantly less than mice (PMID: 35332327) so it would be important to check the ability of CARTs to activate the immune system. Upon strong immune activation, application of CARTs in humans is questionable due to potential toxicity issues.

As suggested by the reviewer, we have now measured the levels of inflammatory cytokines 24h after in vivo delivery of mRNA with bAC CARTs. We found no increase in cytokine levels after i.v. delivery of mRNA complexed with bAC CART. A complete report of cytokine levels is shown in Supplementary figure 14c.

Inflammatory cytokines (Figure S14c):

All changes described here have also been highlighted in the main manuscript.

REVIEWERS' COMMENTS

Reviewer #2 (Remarks to the Author):

Now acceptable as it is.

Reviewer #3 (Remarks to the Author):

The authors appropriately addressed all the concerns of reviewer 3.

As asked by the editor, reviewer 3 also evaluated the answers that were given to address the concerns of reviewer 1. The authors gave adequate answers to all questions/requests of reviewer 1.

Overall, reviewers 1 and 3 have no more concerns about the publication of this interesting manuscript.